# Proteasome inhibitor-induced modulation reveals the spliceosome as a specific therapeutic vulnerability in multiple myeloma

Hector H. Huang [1], Ian D. Ferguson[1], Alexis M. Thornton[2], Prabhakar Bastola [1], Christine Lam[1], Yu-Hsiu T. Lin [1], Priya Choudhry [1], Margarette C. Mariano[1], Makeba D. Marcoulis[1], Chin Fen Teo[3], Julia Malato[4], Paul J. Phojanakong[4], Thomas G. Martin III[4,5], Jeffrey L. Wolf[4,5], Sandy W. Wong[4,5], Nina Shah[4,5], Byron Hann[4], Angela N. Brooks [2] & Arun P. Wiita [1,4✉]

Enhancing the efficacy of proteasome inhibitors (PI) is a central goal in myeloma therapy. We proposed that signaling-level responses after PI may reveal new mechanisms of action that can be therapeutically exploited. Unbiased phosphoproteomics after treatment with the PI carfilzomib surprisingly demonstrates the most prominent phosphorylation changes on splicing related proteins. Spliceosome modulation is invisible to RNA or protein abundance alone. Transcriptome analysis after PI demonstrates broad-scale intron retention, suggestive of spliceosome interference, as well as specific alternative splicing of protein homeostasis machinery components. These findings lead us to evaluate direct spliceosome inhibition in myeloma, which synergizes with carfilzomib and shows potent anti-tumor activity. Functional genomics and exome sequencing further support the spliceosome as a specific vulnerability in myeloma. Our results propose splicing interference as an unrecognized modality of PI mechanism, reveal additional modes of spliceosome modulation, and suggest spliceosome targeting as a promising therapeutic strategy in myeloma.

[1] Department of Laboratory Medicine, University of California, San Francisco, CA, USA. [2] Department of Biomolecular Engineering, University of California, Santa Cruz, CA, USA. [3] Department of Physiology, University of California, San Francisco, CA, USA. [4] Helen Diller Family Comprehensive Cancer Center, University of California, San Francisco, CA, USA. [5] Department of Medicine, University of California, San Francisco, CA, USA. ✉email: Arun.wiita@ucsf.edu

Multiple myeloma (MM) is a clonal malignancy of plasma cells with no known cure. Like normal plasma cells, myeloma cells produce and secrete extremely high amounts of immunoglobulin. This unique function may be exploited by therapeutically inhibiting the proteasome using the Food and Drug Administration-approved proteasome inhibitors (PIs) bortezomib, carfilzomib (Cfz), and ixazomib. Proteotoxic stress caused by these first-line therapeutic agents has been proposed to induce the apoptotic function of the unfolded protein response (UPR)[1], leading to plasma cell death while largely sparing normal tissues[2,3]. However, despite the appealing simplicity of this mechanism, the canonical UPR is not always strongly induced in myeloma cells by PIs[4] and is unlikely to be the sole mode of PI cytotoxicity in MM. Indeed, many additional mechanisms of action of PIs have also been proposed, ranging from nuclear factor (NF)-κB inhibition to immune microenvironment effects to aberrant recycling of cytosolic amino acids[5,6].

Identifying the full range of PI mechanisms of action remains relevant given that acquired PI resistance is clinically widespread but its origins remain unclear[7,8]. Finding new methods to specifically target PI-resistant disease, or molecules to synergize with PIs to avoid resistance by driving deeper remissions, remains a long-standing goal. As one approach to achieving this goal, we and others have studied the response of malignant plasma cells to PIs using both gene expression and proteomic methods[9–11]. Notably, one of the most prominent features of the cellular response to PIs is the activation of the heat shock response[12]. This mechanism leads to significant induction of cytosolic protein-folding chaperones, possibly to assist in protein refolding and decrease in unfolded protein stress. We and others[9,12,13] have therefore proposed targeting mediators of the heat shock response as potential combination therapies with PIs.

However, one unresolved question is whether proteasome inhibition may carry additional effects on plasma cells that are not revealed by mRNA or protein abundance analysis alone. We hypothesized that additional modalities of response, and thereby new myeloma-relevant therapeutic targets, may be revealed by studying the signaling-level response to PIs with unbiased mass spectrometry-based phosphoproteomics. The large majority of therapy-relevant investigations using this technique have focused on elucidating the effects of kinase inhibitors[14]. However, we reasoned that a significant cellular perturbation such as proteasome inhibition would likely also indirectly perturb kinase and phosphatase signaling in a broad fashion.

Here we use unbiased phosphoproteomics to quantify >5000 phosphopeptides in myeloma cells exposed to the irreversible PI Cfz. Surprisingly, we find the greatest increases in phosphorylation occur in proteins associated with the spliceosome machinery. A link between these processes is invisible at the gene expression level. We further evaluate this link from a mechanistic and therapeutic perspective, finding that PIs lead to specific disruption of normal splicing. We suggest interference of splicing as an additional mechanism of action of PIs not previously explored. Inhibition of splicing has recently become a promising therapeutic strategy in other hematologic malignancies[15]. Our results reveal an intersection of cellular stress and the splicing machinery, which may have broad relevance in biology. Furthermore, we propose the spliceosome as a new and potentially selective therapeutic target in myeloma.

## Results

**PI treatment leads to splicing factor phosphorylation.** We first used unbiased phosphoproteomics to examine the signaling-level response of MM.1s MM cells to Cfz. We chose timepoints across

24 h for analysis based on our prior results demonstrating how the proteomic response to PI evolves over many hours[9]. This is in contrast with most phosphoproteomic studies, examining direct kinase effects on a timescale of minutes[14]. Here we instead consider the indirect effects on phosphorylation induced by PI exposure. Using label-free quantification of immobilized metal affinity chromatography-isolated phosphopeptides, we found that altered phosphorylation signatures were most prominent 24 h after treatment (Fig. 1 and Supplementary Fig. 1). In total, we quantified 5791 phosphosites in at least one technical replicate of the timecourse, with >99% of phosphosites representing Ser or Thr phosphorylation events, as expected using this enrichment technique.

At each timepoint, we simultaneously performed single-end RNA-seq for gene expression (Supplementary Data 2). Figure 1b

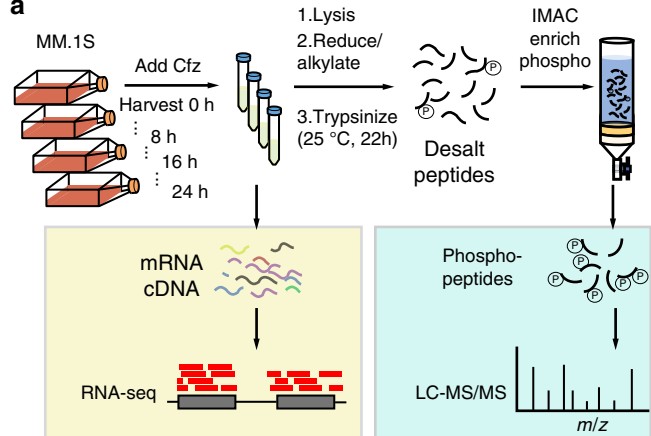

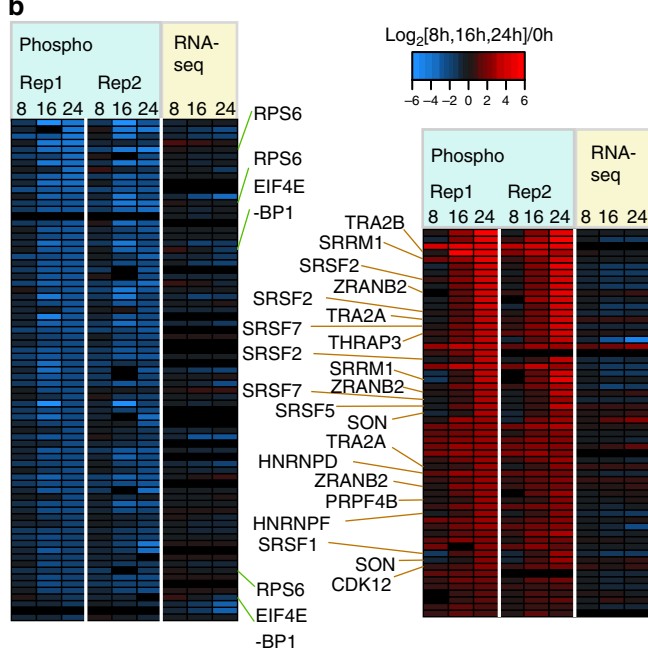

**Fig. 1 Unbiased phosphoproteomic timecourse analysis of MM.1S cells treated with the PI carfilzomib (Cfz). a** Workflow of timecourse treatment of MM.1S cells with Cfz. Cells were allotted for both RNA-seq analysis and LC-MS/MS. **b** Downregulated (left panel) and upregulated (right panel) log₂-transformed phosphopeptide MS1 intensities for two technical replicates of proteins with unchanged transcript levels (RNA-seq). Labels highlight dephosphorylation of RPS6 and EIF4EBP1 on the left and phosphorylation of splicing-related proteins on the right.

shows 58 upregulated (red) and 75 downregulated (blue) phosphopeptides from proteins with largely unchanged RNA transcript abundance as detected by unsupervised hierarchical clustering. We found decreased phosphorylation of the translation factor EIF4E-BP1 as well as RPS6 (Fig. 1b), as expected upon PI-induced cellular stress[9]. While other downregulated phosphopeptides did not suggest any highly enriched biological function, among upregulated phosphosites we were surprised to find that 14 of the 58 were present on proteins related to pre-mRNA splicing. These primarily included the heterogeneous ribonucleoprotein (HNRNP) family as well as SRSF splicing factors (Fig. 1b). In particular, the arginine- and serine-rich "RS" domain of the SRSF proteins are known to have their activity modulated by phosphorylation[16]. Notably, these prominent signaling-level effects on splicing factors were invisible to prior gene expression studies of PI response and had not been investigated. We therefore chose to further explore the interaction between PIs and the splicing machinery.

To validate this initial result from label-free proteomics, we used a stable isotope labeling by amino acids in cell culture (SILAC) phosphoproteomics approach at the 24-h timepoint. We evaluated both a low dose (10 nM, $n = 2$ biological replicates) and a moderate dose (18 nM, $n = 2$) of Cfz (Fig. 2a, b). Of the 520 phosphosites significantly upregulated ($p < 0.05$; ≥2-fold-change) in MM.1S treated with 18 nM Cfz in Fig. 2a, 127 (24.4%) are associated with splicing-related proteins, with 23 of these as part of the SRSF protein family of splicing factors. Background-

corrected Gene Ontology (GO) analysis confirms that all of the top enriched biological processes involve RNA splicing regulation and mRNA processing (Fig. 2e and Supplementary Fig. 1b). This signaling response is much weaker at 10 nM Cfz, with only 25 upregulated phosphosites and none that are splicing related. These results suggest a strong dose–response effect of phosphorylation changes after proteasome inhibition, both across splicing factors and the broader proteome.

To differentiate changes at the signaling level to changes at the protein level, unenriched peptides were also analyzed (Supplementary Fig. 2a, b). Confirming expected responses, the most upregulated proteins included heat shock-induced chaperones as well as SQSTM1/p62 (ref. [9]). In contrast, splicing factors do not significantly change in abundance, confirming that phosphosite increases are due to changes at the signaling level and not protein copy number.

**Melphalan induces similar but not identical phosphorylation**. We next investigated whether this broad splicing factor phosphorylation phenotype was unique to PI. We compared the MM.1S response to 10 μM melphalan, a DNA alkylating agent and clinically used myeloma therapeutic, again by SILAC-labeled phosphoproteomics at 24 h. In parallel, we also treated another MM cell line, AMO-1, with 15 nM Cfz to determine whether the phosphorylation response is consistent across cell line models.

Here both conditions led to ~20% cell death (Supplementary Fig. 3a). Western blot confirmed induction of DNA damage by

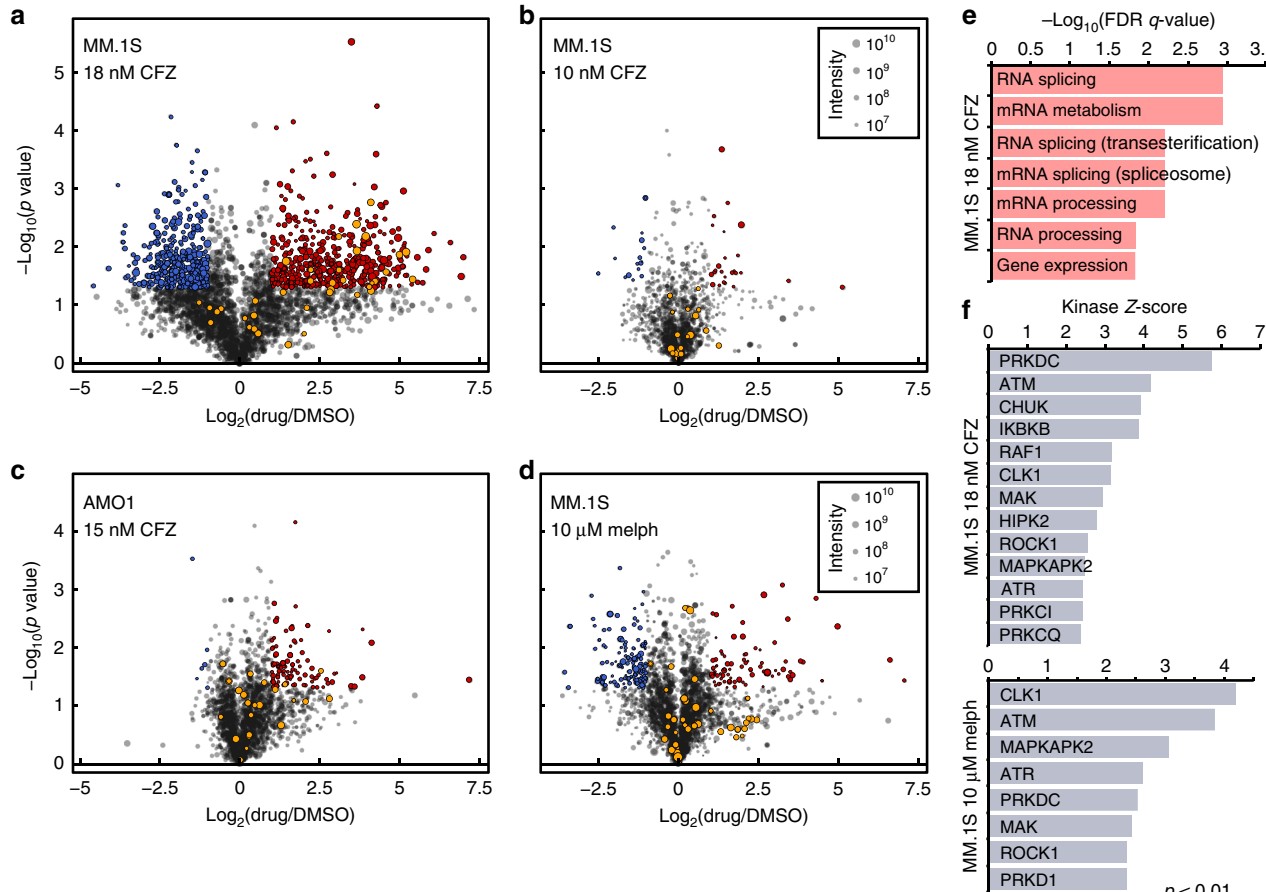

**Fig. 2 Cfz induces phosphorylation of splicing factors in a dose-responsive manner. a–d** Volcano plots of log2-transformed ratios of phosphosite abundances between **a** MM.1S treated with 18 nM Cfz, **b** 10 nM Cfz, or **d** 10 μM melphalan compared to DMSO and **c** AMO-1 with 15 nM Cfz compared to DMSO. Significant upregulated sites are in red, while downregulated are in blue (>2-fold change, $p < 0.05$). SRSF-related sites are in orange. Circle size corresponds to summed SILAC intensities. **e** Top-ranked GO terms for genes with significantly upregulated phosphosites in MM.1S cells treated with 18 nM Cfz. **f** Top ranked KSEA-activated kinases (with at least 5 substrates) for MM.1S treated with 18 nM Cfz (top panel) and 10 μM melphalan (bottom panel).

melphalan and proteotoxic stress response for Cfz (Supplementary Fig. 3c, d). Compared to 18 nM Cfz, we saw largely decreased phosphorylation-level responses to both of these agents (Fig. 2c, d). Of the 113 phosphosites significantly upregulated in AMO-1, 7 belong to splicing-related proteins (SRSF2, SRSF6, SRRM1, HNRNPH1, TRA2A, DDX1). This result is consistent with the MM.1S results in Fig. 2a, b, where greater PI response correlates with more prominent phosphorylation changes.

Under 10 μM melphalan, 93 phosphosites were significantly upregulated, with 8 sites on splicing-related proteins (HNRNPK, TRA2A, SRRM2, and WDR77), but not SRSF proteins (Fig. 2d). As expected, both shotgun proteomics and RNA-seq for gene expression confirm that PI and DNA damage elicit different responses (Supplementary Fig. 2a–e).

We further performed Kinase Substrate Enrichment Analysis[17] to identify kinases whose activity may regulate differential phosphorylation found by phosphoproteomics. While this tool is limited by its reliance on well-characterized kinase–substrate relationships, within this framework, this tool identified similar kinases active under both 18 nM Cfz and 10 μM melphalan treatment (Fig. 2f). Notably, both drugs are predicted to induce cdc2-like kinase 1 (CLK1), a kinase known to phosphorylate SRSFs among other proteins[18]. Furthermore, the increased activity of DNA-dependent protein kinase and ATM kinase are consistent with apoptosis-induced activation[19]. However, in line with PI biology, Cfz also strongly induced inhibitory kappa B kinase activity, a kinase leading to NF-κB inhibition after PI treatment[20]. Taken together, these results indicate that drug-induced stress may broadly lead to phosphorylation of splicing factors, though precise patterns of phosphorylation may differ in a drug-specific manner.

**SRSF proteins appear highly phosphorylated at baseline**. To investigate our phosphoproteomic results via an orthogonal method, we performed western blots to evaluate for phosphorylation-induced gel mobility shift of SRSF1, SRSF3, and SRSF6. While we initially saw no discernable alteration after Cfz, treatment with calf intestinal phosphatase resulted in a substantial shift of SRSF proteins but not actin (Supplementary Fig. 3f). Therefore, these factors exist in a highly phosphorylated state even at baseline in MM cells. While the mass spectrometry-detected changes at specific phosphosites may still result in biological effects, Cfz-induced modulation does not appear to reflect a dramatic shift in the overall phosphorylation status of these SRSF proteins in this system.

To further investigate baseline phosphorylation status of SRSF proteins, we treated MM.1S cells with 50 μM KH-CB19 (ref. [21]), a reportedly highly selective inhibitor of the SRSF kinases CLK1 and CLK4 ($K_D = 20$ nM vs. CLK1). Unbiased phosphoproteomics after 24 h surprisingly showed no significant change in phosphorylation status of any quantified SRSF phosphosites, except one upregulated (Supplementary Fig. 2f).

**Proteasome inhibition induces intron retention (IR) in MM cells**. We next investigated whether pre-mRNA splicing itself was altered after drug treatment. We obtained paired-end sequencing data from polyA-enriched RNA on the same samples used for phosphoproteomics, plus one additional biological replicate ($n = 3$ total) of each of the following: MM.1S treated with 18 nM Cfz, with 10 μM melphalan, and with dimethyl sulfoxide (DMSO) as control; and AMO-1 treated with 15 nM Cfz and with DMSO as control. We used JuncBASE[22] to process the aligned sequencing data by identifying and quantifying both annotated and novel splice junctions. Data for each alternative splicing event was evaluated using the standard measure of "percent spliced in" or PSI (Fig. 3a).

Comparative analysis of differential PSI (ΔPSI) between 18 nM Cfz- and DMSO-treated MM.1S was performed across several alternative splicing outcomes (Fig. 3b, Supplementary Data 3). The ΔPSI distribution for IR demonstrated the greatest positive shift after Cfz ($n = 25,807$ total IR events measured; median = 2.54). ΔPSI medians for alternative splice site selection also demonstrated a significant shift (alt. donor median = 1.64, $n = 9188$ and alt. acceptor median = 0.81, $n = 9352$). All other categories were closer to a median of zero ($p < 2.2E{-}16$ for median of IR distribution, alt. donor, and alt. acceptor vs. median of cassette by Mann–Whitney test, Supplementary Data 3). Intriguingly, PIs are well known to induce a strong heat shock response[12]. Prior work in non-cancer cells demonstrated that heat shock alone could impair splicing and induce IR without broadly affecting other alternative splicing events[23,24]. In general, intron-retained transcripts may be subject to nonsense-mediated decay or retained in the nucleus where they remain untranslated. Our results suggest that a similar splicing impairment may be present in MM cells exposed to PI.

We then considered the possibility that the IR phenotype results from a global dysfunction of the splicing machinery during drug-induced apoptosis, which is likely occurring with ~85% cell death at our high-dose Cfz treatment (Supplementary Fig. 3a). Our prior data indicated that SF3B1 and U2AF2, core components of the splicing machinery, are some of the earliest substrates cleaved by caspases during PI-induced apoptosis[9]. Indeed, we validated by western blotting that SF3B1 and U2AF2 are proteolytically cleaved after Cfz treatment, and this cleavage can be blocked by the pan-caspase inhibitor zVAD-fmk (Supplementary Fig. 3e).

We therefore considered whether these caspase cleavage events could be responsible for the IR phenomenon. However, we found similar IR shift in AMO-1 cells treated with 15 nM Cfz ($n = 27,386$; median = 2.2) (Fig. 3c) despite much less cytotoxicity (~20%) than 18 nM Cfz in MM.1S. As caspase cleavage correlates with degree of cell death, it therefore appears unlikely that cytotoxicity alone is responsible for IR. Notably, an even smaller shift was observed in IR for MM.1S with 10 μM melphalan, also at ~20% cytotoxicity ($n = 24,247$; median = 0.44; $p < 2.2E{-}16$ for IR distribution MM.1S 18 nM Cfz vs. 10 μM melphalan). Instead, after treatment with melphalan, the greatest ΔPSI shift occurred with alt. exon cassettes (single cassette median = 1.06, $n = 12,267$, coordinated cassette median = 1.75, $n = 1417$, Fig. 3d). Even if we only consider statistically significant IR events ($p < 0.05$), the drug responses remain distinct (Supplementary Fig. 4a, b). Therefore, while melphalan also affects alternative splicing, it appears to do so via a different mechanism than Cfz[25] (Fig. 3e).

**Overexpressed SRSF1 variants do not alter splicing**. We next considered whether PI-induced splicing factor phosphorylation and the IR signature are causally linked. To investigate this question, we considered SRSF1 (also known as SF2 or ASF), a well-characterized splicing factor and a putative proto-oncogene[16,26]. We found that SRSF1 demonstrates upregulated phosphorylation within the arginine- and serine-rich RS1 and RS2 domains after Cfz (Fig. 2a, Supplementary Data 1). The current model of SRSF1 function suggests that (1) SR protein kinase (SRPK)-mediated phosphorylation of RS domain leads to translocation into the nucleus, (2) further hyperphosphorylation by CLK1 causes association with the U1 spliceosome, and (3) partial dephosphorylation is required for splicing catalysis[16,27,28].

To study the effects of SRSF1 phosphorylation, we exogenously expressed a wild type (SRSF1-WT), phosphomimetic (SRSF1-SD),

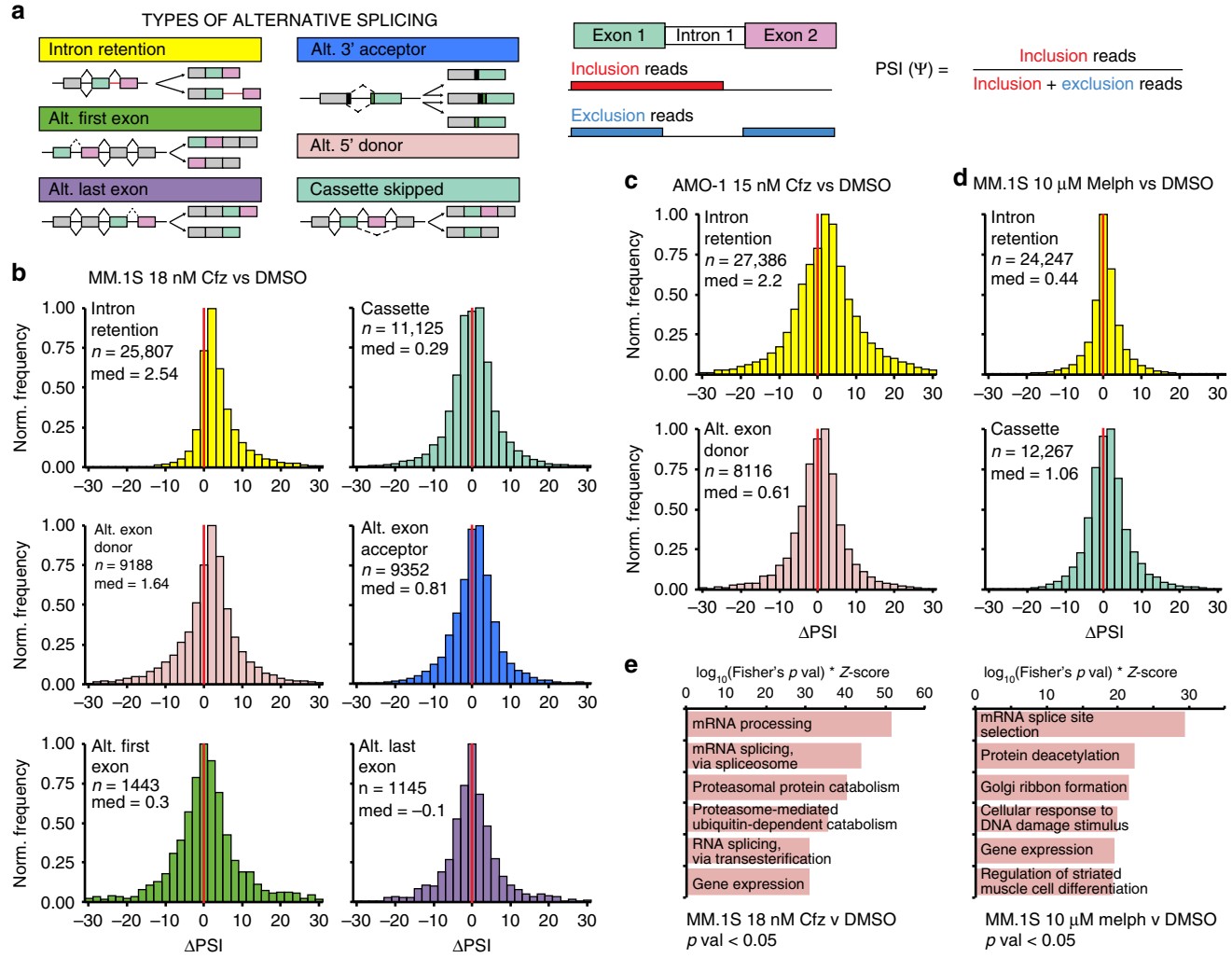

**Fig. 3 Cfz treatment leads to prominent intron retention. a** Cartoon description of alternative splicing event (ASE) types and description of ΔPSI.
**b** Histograms of ΔPSI for JuncBASE identified ASEs in MM.1S treated with 18 nM Cfz stratified according to type of splicing event (IR = yellow, alt. exon cassette = teal, alt. exon donor (5′ splice site) = pink, alt. exon acceptor (3′ splice site) = blue, alt. first exon = green, alt. last exon = purple). Bin = 2, red line indicates ΔPSI = 0. **c** Histograms of ΔPSI for all IR events (top panel) and Alt. exon donor events (bottom panel) in AMO-1 treated with 15 nM Cfz.
**d** Histograms of ΔPSI for all IR events (top panel) and Alt. cassette events (bottom panel) in MM.1S treated with 10 μM melphalan. **e** Top ranked GO enrichment terms for genes with significant (p < 0.05) ASEs for MM.1S cells treated with 18 nM Cfz (left panel) or with 10 μM melphalan (right panel).

or phosphodead (SRSF1-SA) variant in AMO-1 cells. We assumed an all-or-none model, where exogenous SRSF1 mutants have all 20 serines in the RS1 and RS2 domains replaced with either an aspartate (SD) or an alanine (SA) (Fig. 4a). Phosphorylation of the RS1 domain is thought to be necessary for nuclear localization[29,30]; the attempted forced nuclear localization of the SA mutant via a nuclear localization signal (NLS) was chosen to probe potential splicing-level effects of phosphodead SRSF1 interacting with the spliceosome. Immunoblot confirmed expression of exogenous SRSF1 constructs at lower levels than the high-abundance endogenous protein (Supplementary Fig. 4e). Epi-fluorescent images of these mCherry fusions (Fig. 4b) show that most of WT and SD signal is localized to the nucleus, consistent with expected biology. However, a much larger fraction of SA mutant is trapped in the cytosol despite NLS tagging. Consistent with prior work[31,32], this finding suggests that RS phosphorylation is a major requirement for entry into the nucleus.

Upon JuncBASE analysis of untreated polyA RNA-seq data (n = 3 for each SRSF1 variant), we saw remarkably few global differences in PSI as a function of modeled SRSF1 phosphorylation status (Fig. 4d). Notably, our results in Fig. 1b suggested that

phosphorylation of multiple splicing factors, including other SRSF proteins, occurs simultaneously under Cfz-induced stress; altered phosphorylation of SRSF1, alone, may not carry any significant effects.

**SRSF1 phosphomimetic has weakened spliceosome interaction.** Though we cannot draw a direct link between SRSF1 phosphorylation status and specific alternative splicing events, we further investigated the diverse biological roles of SRSF1. These include regulating nuclear export of spliced mRNAs and translational regulation in the cytosol[33–35]. Using 3×-FLAG tag on our SRSF1 constructs, we performed affinity purification mass spectrometry (AP-MS) with label-free quantitative proteomics vs. an mCherry-NLS-[FLAG]₃ control.

While clear differences were observed between the nuclear and cytosolic interactome for each construct, overall GO signatures were surprisingly similar across WT, SD, and SA within each compartment (Supplementary Fig. 5b, d, f). In the cytosol, we found consistent interactions between both SRSF1-WT and -SD with several RNA-binding proteins as well as components of the

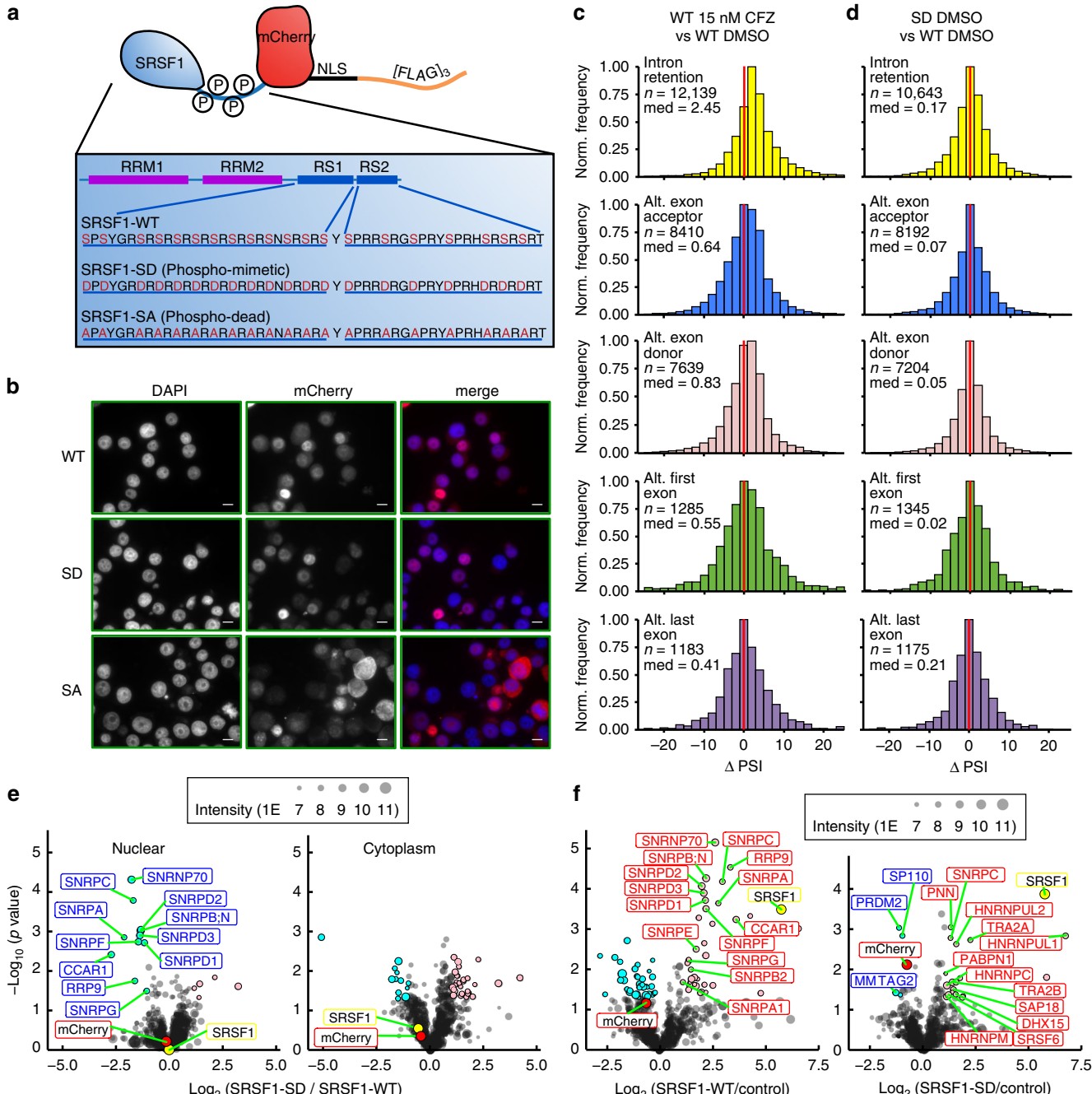

**Fig. 4 Modeling SRSF1 phosphorylation in MM drives interactome dynamics but not global splicing changes. a** Cartoon of protein architecture for SRSF1-NLS-mCherry-[FLAG]$_3$. **b** Epi-fluorescent images of DAPI-stained AMO-1-expressing mCherry-labeled SRSF1-WT (top panels), SRSF1-SD (middle panels), and SRSF1-SA (bottom panels). Scale bar represents 10 μm, DAPI stain is pseudo-colored in blue and mCherry is pseudo-colored in red in the merged image, and micrographs represent experiments repeated twice with similar results. **c, d** Histograms of ΔPSI for IR, alt. exon donor, alt. exon acceptor, alt. first exon, and alt. last exon ASEs when comparing differential splicing of AMO-1-expressing **c** SRSF1-WT treated with 15 nM Cfz to DMSO and **d** SD to WT. **e** Volcano plots indicating differential interactors of SD compared to WT in both the nucleus and cytoplasm. **f** Volcano plots of WT or SD when compared to control (NLS-mCherry-[FLAG]$_3$) in AMO-1 nucleus reveal SD exclusion from spliceosome. Significant enriched proteins in pink, unenriched proteins in cyan ($p < 0.05$, ≥2-fold change). Circle size corresponds to summed LFQ intensities. mCherry ratio is red and SRSF1 ratio is yellow.

translational machinery. We do note one stark difference between WT and SD mutant in the nuclear fraction: only the WT construct showed interaction with several small nuclear ribonucleoproteins, core components of the U1–U2 spliceosome (Fig. 4e). Unexpectedly, these nuclear interactions were not enriched in the SD construct, which instead interacted with other splicing-related factors, such as TRA2A, TRA2B, and PABPN (Fig. 4f). This interactome mapping may help refine the current model of SRSF1 biology, which suggests that hyperphosphorylation of RS domains leads to preferential integration with the U1 spliceosome[36,37].

**PI leads to both IR and specific exon usage.** We next explored the splicing-level effects of 15 nM Cfz treatment on these SRSF1-modified cells. Notably, in this setting cytotoxicity at 24 h was <10% in 8 of the 9 total replicates (Supplementary Fig. 4d). Cfz

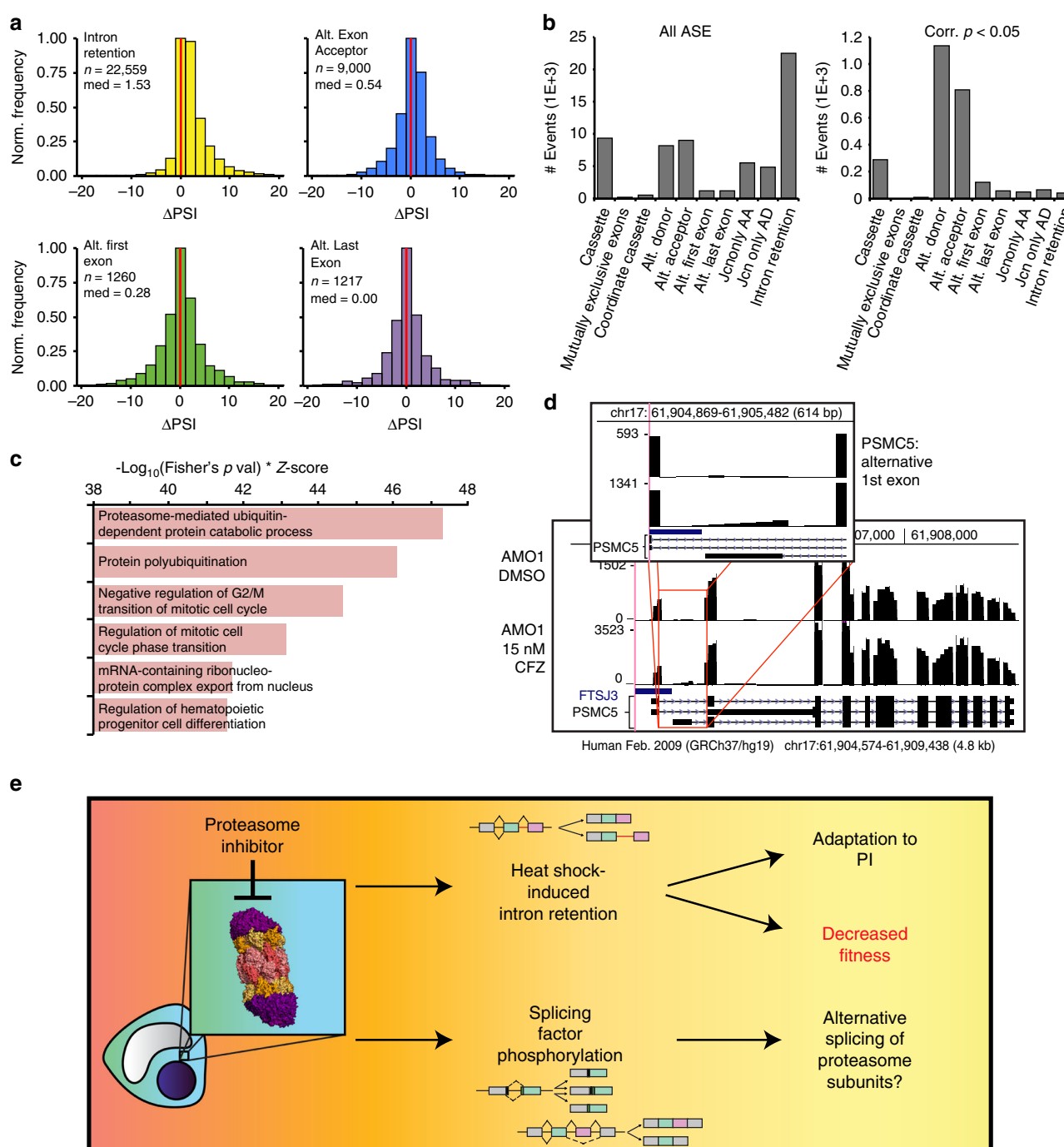

**Fig. 5 Combined SRSF1 constructs validate the splicing phenotype after Cfz. a** Histograms of ΔPSI in pooled analysis of parental AMO−1-, SRSF1-WT-, SD-, and SA-expressing cells treated with 15 nM Cfz compared to DMSO. **b** Graph shows total number of events ($n = 62,474$) for each ASE type (left panel) and only the significant (FDR-corrected $p < 0.05$) events ($n = 2,575$) for each type (right panel). **c** Top ranked GO enrichment terms of all genes involved in significant ASEs, regardless of type. **d** RNA-seq coverage map (UCSC Genome Browser) compares counts for the proteasomal subunit *PSMC5* between AMO-1 treated with 15 nM Cfz (bottom panel) and with DMSO (top panel). Inset displays sequencing counts showing alternative first exon. **e** Model of new PI mechanism of action found in MM.

again elicited a response consistent with that found in Fig. 3c: despite minimal cell death, we observed a clear shift in the median ΔPSI toward increased global IR ($n = 12,139$; median = 2.45, $p < 2.2E−16$ for one-sample, two-tailed Wilcoxon summed rank test; Fig. 4c). These findings in the absence of apoptosis underscore that caspase cleavage of splicing factors is unlikely to be a primary mechanism of IR after PI.

The combined RNA-seq dataset of all Cfz-treated samples were analyzed together (Fig. 5a; $n = 24$ total replicates across all AMO-1, including data in Figs. 3c and 4c). However, despite detecting $n = 22,559$ IR events by JuncBASE (Fig. 5b, left panel; example in Supplementary Fig. 4g), very few individual transcripts ($n = 43$) showed statistically significant (false discovery rate-corrected $p < 0.05$) IR across replicates (Fig. 5b, right panel). This finding

suggests that Cfz-induced IR may be a stochastic process, perhaps resulting from general interference with the splicing machinery without a coherent selection for specific transcripts.

In contrast, alternative exon splice site usage (exon donor ($n = 1134$) and acceptor ($n = 810$)) emerged as the dominant type of alternative splicing when considering only statistically significant events (Fig. 5b). Interestingly, GO analysis (Supplementary Data 4) of all genes undergoing significant alternative splicing after Cfz ($n = 2575$ events in Fig. 5b, right panel) revealed "proteasome-mediated ubiquitin-dependent protein catabolic process" ($p = 2.08E{-}16$) and "protein polyubiquitination" ($p = 1.39E{-}13$) as highly enriched (Fig. 5c). Multiple proteasome subunits (*PSMA3/4/5/7*, *PSMB4/5*, *PSMC1/4/5*, *PSMD1-4*, *PSME2*), the protein homeostasis node p97 (*VCP*), and ubiquitin (*UBB*, *UBC*) all undergo some degree of alternative splicing after Cfz (example in Fig. 5d). These findings raise the possibility that alternative splicing may modulate the protein homeostasis machinery in response to PI.

We further investigated potential mechanisms by which PIs may specifically alter splicing. First, we treated MM.1S cells with 5 nM bortezomib, a reversible PI and first-line MM therapy. While we again noted widespread phosphorylation of SR splicing factors (Supplementary Fig. 6a), we surprisingly did not observe a global increase in IR (Supplementary Fig. 7a). However, we again confirmed specific enrichment of alternative splicing patterns in the ubiquitin–proteasome system components, suggesting these alterations are part of a PI-specific biologic response to splicing (Supplementary Fig. 7d).

In contrast, treatment with 50 μM lenalidomide led to no change in phosphorylation patterns (Supplementary Fig. 6b). Combination experiments with Cfz and TG003, an alternate CLK1/4 inhibitor[38], vs. Cfz alone, showed increased phosphorylation of SR domains after 24 h of treatment, consistent with rebound phosphorylation of substrates under sustained CLK inhibition[39] (Supplementary Fig. 6c). We further combined z-VAD-FMK with Cfz and found that, remarkably, caspase blockade largely abrogated the phosphorylation phenotype (Supplementary Fig. 6d). This result is consistent with our Cfz dose-dependent findings, which showed that phosphorylation corresponded with cell death (Fig. 2).

We next investigated the hypothesis that the IR phenotype we observed after Cfz could be related to PI-induced heat shock. The leading proposed mechanism for heat shock-induced alternative splicing, first noted over 30 years ago[23], is that dephosphorylation of SRSF10/SRp38 plays an important role[40]. In our system, SRSF10 immunoblotting did not show any clear alteration in phosphorylation after Cfz-induced heat shock response (Supplementary Fig. 3g). However, we note a loss of the full-length SRSF10 band at high levels of cell death, suggesting that this protein is likely a caspase substrate. Therefore, loss of SRSF10 may play a role in this biological system, but it also appears unlikely to mediate the IR phenotype at low Cfz doses incurring minimal cell death.

We also investigated the hypothesis that SRSF1 localization dynamics may affect exon selection dynamics after either Cfz or heat shock. Treatment with Cfz or heat shock did not cause any re-distribution to the cytosol of SRSF1 WT-mCherry by confocal microscopy (Supplementary Fig. 8). However, we noticed a trend toward SRSF1 redistribution in the nucleus after Cfz treatment (Supplementary Fig. 8e–h), which could play a role in splicing alterations.

While not yet definitive, our work offers a preliminary model for the effects of PI on the splicing machinery in myeloma (Fig. 5e). Upon therapeutic insult, the stress response induces phosphorylation of multiple splicing factors, potentially leading to specific alterations in exon usage based on known SRSF biochemistry.

Modification of the proteasome itself via alternative splicing may also play a role in adaptation or resistance to PI. In parallel, we observe an increase in stochastically distributed IR events. These events, typically leading to non-functional transcripts, may work to reduce proteotoxic stress and conserve cellular resources normally devoted to protein synthesis, thereby playing a role in adaptation to proteasome inhibition. Alternatively, the IR phenotype may indicate malfunction of the spliceosome, an essential process whose loss reduces tumor cell fitness. Interference with splicing may therefore be a previously unappreciated part of the Cfz mechanism of action.

**E7107 is broadly potent vs. MM and synergistic with PI.** Expanding on this relationship between PI and splicing interference, we further investigated the therapeutic potential of more dramatic spliceosome disruption in myeloma. We employed the tool compound E7107, a direct inhibitor of the core U2 catalytic spliceosome component SF3B1[15]. This molecule is known to induce extreme IR and strong cytotoxic effects in models of myeloid malignancy[41].

Using both quantitative PCR validation of canonical IR events after SF3B1 inhibition[42] as well as JuncBASE analysis of RNA-seq data (Fig. 6a), as expected, we identified marked IR after 6 h of 10 nM E7107 in MM.1S cells (ΔPSI median = 13.79, $n = 30{,}666$). The number of significant ($p < 0.05$) IR events remained very high with E7107 ($n = 7171$), unlike the apparently stochastic IR events seen with PI (Supplementary Fig. 9c). Furthermore, unlike with PI, we observed massive global loss of cassette exon splicing (ΔPSI median = −16.6, $n = 24{,}053$, Supplementary Fig. 9d). Altogether, this suggests that PI-induced impairment of splicing is a partial interference of normal splicing, unlike the total abrogation seen with E7107. Underscoring the potential of splicing inhibition as a therapeutic strategy in MM, E7107 was extremely potent vs. a panel of seven MM cell lines treated for 48 h, with $LC_{50}$s ranging from <1 nM to 30 nM (Fig. 6b). In addition, a PI-resistant AMO-1 cell line[43] showed similar sensitivity to E7107 as the parental line (Fig. 6c). This finding suggests the potential for clinical utility of splicing inhibition even in PI-refractory disease.

We noted that our MM cell line sensitivities appeared essentially bimodal, with one group of more sensitive lines with $LC_{50}$s of <1 nM and another slightly less sensitive group of cell lines with $LC_{50}$ of 20–50 nM. Examining publicly available MM cell line gene expression data (www.keatslab.org), we were intrigued to find that the more sensitive lines demonstrated significantly higher RNA expression of *SF3B1* (Fig. 6d). Unfortunately, though, this result was not confirmed at the protein level (Fig. 6e). We found no other clear candidates for markers of E7107 sensitivity based on available DNA or RNA sequencing data from this limited cohort of cell lines. Baseline comparison of alternative splicing patterns in the sensitive AMO-1 vs. insensitive MM.1S showed no global shifts in splicing patterns (Supplementary Fig. 7c).

We further explored the hypothesis that interfering with splicing via two different mechanisms may lead to synergistic MM cell death. Indeed, combination studies with Cfz and E7107 showed strong synergy across the dosing landscape based on ZIP synergy scoring[44] (Fig. 6f). In contrast, melphalan, which induced much less IR than PI (Fig. 3), showed weak antagonism in combination with E7107 (Fig. 6g). Bortezomib, which induced minimal IR (Supplementary Fig. 7a), also showed weaker synergy with E7107 than with Cfz (Fig. 6i). Recently, it has been proposed that PI resistance can be overcome by targeting mitochondrial biology[45,46]. Interestingly, venetoclax, which targets mitochondria to induce apoptosis, also strongly synergized with E7107,

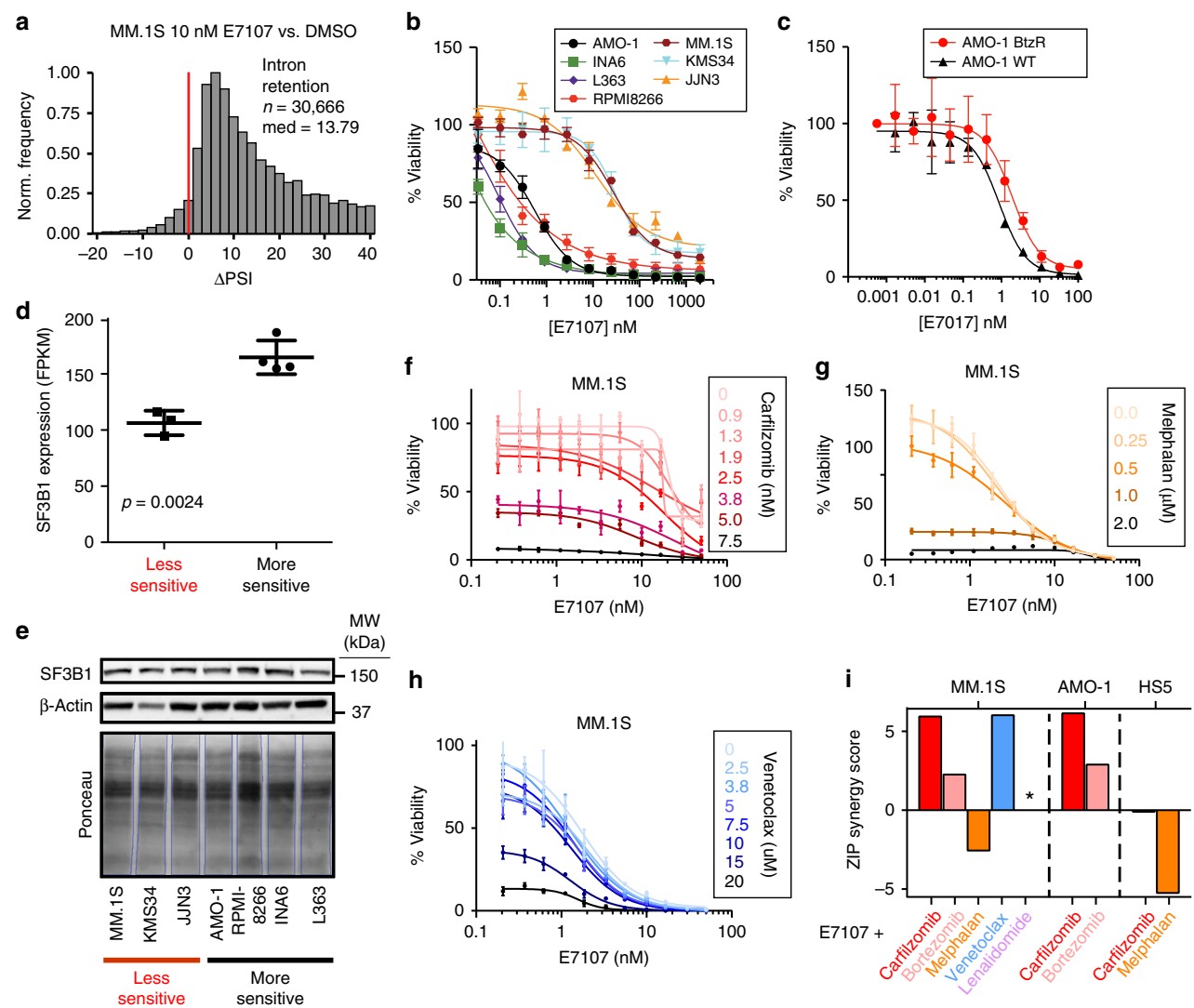

**Fig. 6 The catalytic spliceosome inhibitor E7107 induces IR and has potent anti-MM activity in vitro. a** ΔPSI histogram of IR events for MM.1S treated with 10 nM E7107 for 6 h with respect to DMSO. Bin = 2 and red line at ΔPSI = 0. **b** Cell viability curves compare a panel of 7 MM cell lines' sensitivity to E7107 for 48 h ($n = 4$ technical replicates; mean +/− S.D.) and **c** parental (WT) and PI-resistant (BtzR) AMO-1 cell lines with E7107 ($n = 4$ technical replicates; mean +/− S.D.) **d, e** Evaluation of SF3B1 expression by **d** RNA-seq (www.keatslab.org, mean FPKM +/− S.D.) and **e** western blot across more (AMO-1, INA6, L363, RPMI8266) and less (MM.1S, KMS34, JJN3) E7107-sensitive cell lines. Statistical significance in **d** determined by two-sided $t$ test between ($n = 4$) more sensitive cell lines and ($n = 3$) less sensitive lines, and western blot in **e** was repeated once with similar results. Source data are provided as a Source Data file. **f–h** Cell viability curves of MM.1S combination therapy with E7107 and **f** carfilzomib, **g** melphalan, and **h** venetoclax ($n = 4$; mean +/− S.D.). **i** ZIP synergy score from E7107 combination study. Asterisk (*) denotes no viability effect and zip score is not calculated (see Supplementary Fig. 10b).

potentially indicating some cross-talk between these mechanisms of anti-MM action (Fig. 6h, i). Notably, E7107 and Cfz showed similar amount of synergy in both MM.1S and AMO-1 cells (Fig. 6i, Supplementary Fig. 10e, f) but did not show any synergy in HS5 bone marrow (BM) stromal cells (Fig. 6i, Supplementary Fig. 10c, d), suggesting some potential for specificity for this combination in plasma cells. These findings support the approach of using splicing inhibitors in combination with Cfz in MM treatment. Also, these results strengthen the hypothesis that splicing interference is a part of the Cfz mechanism of action.

**E7107 is highly potent both in vivo and ex vivo.** Based on this encouraging in vitro data, we moved into a standard in vivo MM model of luciferase-labeled MM.1S cells implanted intravenously into NOD *scid* gamma (NSG) mice. Tumor cells home to murine

BM, partially recapitulating the tumor microenvironment in human disease[47]. We found that E7107 was generally well tolerated with no appreciable weight loss (Supplementary Fig. 11a). At 3 mg kg$^{-1}$ E7107 intravenous, a relatively low dose compared to prior studies in other malignancies[41], we still found pronounced anti-MM effect after a brief 2-week treatment (Fig. 7a–c). This suppression of tumor translated into a significant survival benefit ($p = 0.01$, log-ranked test; $n = 6$ per arm).

Next, fresh BM mononuclear cells from seven PI-refractory MM patients were treated for 48 h with varying doses of E7107. Based on flow cytometry of CD138+ plasma cells (Supplementary Fig. 11b, c), we found similar sensitivity of patient tumor cells to E7107 as found in cell lines, with estimated LC$_{50}$s in the low-nM range (Fig. 7d). Notably, CD138-negative BM mononuclear cells showed remarkably little cytotoxicity at these same doses,

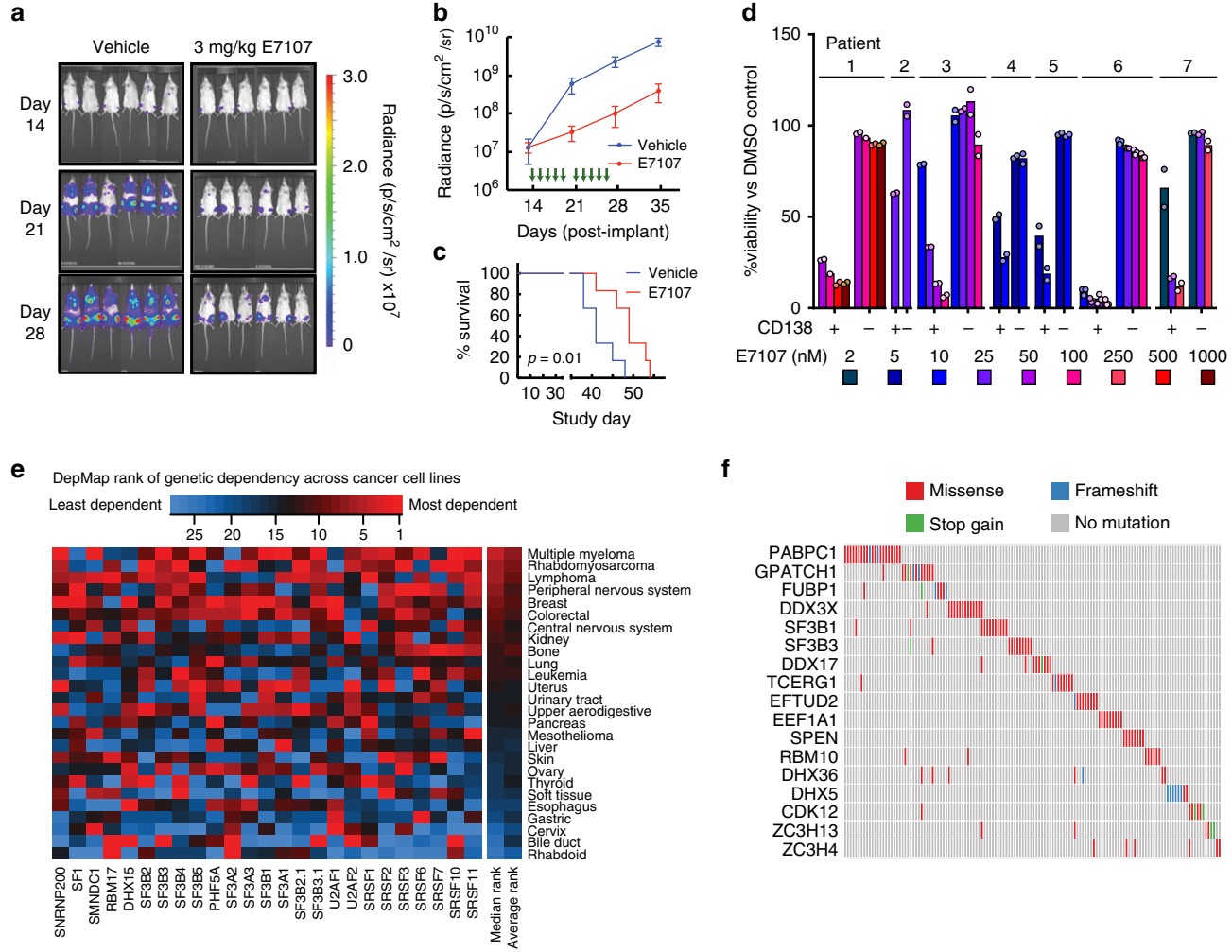

**Fig. 7 Inhibition of the spliceosome is a promising therapeutic strategy in myeloma. a** Bioluminescence imaging of luciferase-labeled MM.1S cells implanted in mice treated with either vehicle (left panel, $n = 6$) or 3 mg kg$^{-1}$ E7107 (right panel, $n = 6$). **b** Mean $+/-$ S.D. luciferase intensity shows lower tumor burden in E7107-treated mice. Green arrows indicate drug administration days (14–18, 21–25). **c** E7107 leads to significant improvement in murine survival ($p = 0.01$ by two-sided log-ranked test). **d** Treatment of primary bone marrow aspirate samples from PI-refractory myeloma patients at various doses of E7107 for 24 h shows significant cytotoxicity to CD138 + MM plasma cells at low-nM concentrations but minimal effects on other (CD138−) hematopoietic cells ($n = 2$ technical replicates; mean is represented by bars). **e** Heatmap of CRISPR-Cas9 essentiality screen data analysis in the Cancer Dependency Map (www.depmap.org; Avana 18Q4 release) of core spliceosomal subunits among all tested tumor cell types. **f** Analysis of MMRF CoMMpass data (research.themmrf.org; release IA11) summarizing mutations with possible functional effects in numerous splicing-related factors, as defined by Seiler et al.[52], within MM patient plasma cells.

supporting a potential therapeutic index for splicing inhibitors in MM.

**Genomics suggests clinical application of splicing inhibitor.** Analysis of CRISPR essentiality screen data in the Cancer Dependency Map (www.depmap.org; Avana library public 18Q4[48]), across >400 cancer cell lines, demonstrated that myeloma has among the strongest genetic dependencies on the E7107 target *SF3B1* (Supplementary Fig. 11d). This genetic data further support the ability to pharmacologically eliminate MM cells via splicing inhibition while sparing normal cells. We extended this analysis to other core components of the U1-U2 spliceosome found to be "common essential" genes per DepMap[49]. We found that MM lines are the most sensitive tumor cell type to genetic ablation of these components, necessary for association with pre-mRNA and splicing catalysis (Fig. 7e). Compared to essential subunits of the 20S proteasome (including the direct PI target *PSMB5*) (Supplementary Fig. 11e), we surprisingly found more

favorable genetic evidence for targeting the spliceosome than the proteasome in MM.

Furthermore, a recent study validated the sulfonamide indisulam as an inhibitor of splicing via targeted degradation of RBM39[50]. We confirmed cytotoxicity of indisulam vs. a panel of MM cell lines (Supplementary Fig. 11f), although LC$_{50}$s (0.3–20 μM) were much higher than those for the SF3B1 inhibitor E7107. In DepMap, MM was again among the more sensitive tumor types to RBM39 ablation (Supplementary Fig. 11g). Indisulam may therefore represent another approach to targeting the spliceosome in MM.

We next took advantage of genomic and transcriptomic data from newly diagnosed MM patient tumors in the Multiple Myeloma Research Foundation CoMMpass study (research. themmrf.org; version IA11). First evaluating gene expression data, we found significantly decreased progression-free survival among patients in the top quartile of *SRSF1* expression vs. those in the bottom quartile ($p = 0.0081$ by log-ranked test) and a trend toward similarly decreased overall survival for patients in the top vs. bottom quartile of *SF3B1* expression ($p = 0.087$;

Supplementary Fig. 11h). These results raise the possibility of poorer outcomes in patients whose disease is more dependent on the spliceosome.

Both E7107[41] and the recently described splicing inhibitor H3B-8800[42] have the greatest potency vs. hematopoietic malignancies carrying mutations in splicing factors such as *SF3B1*, *SRSF2*, *U2AF2*, and *ZRSR2*[51]. These mutations are seen frequently in myelodysplastic syndromes (MDS), acute myeloid leukemia, and chronic lymphocytic leukemia, appearing in up to 50% of MDS patients[51]. We therefore examined exome sequencing data in CoMMpass and found that 28.0% of MM patients (268 of 956) were found to carry missense mutations within at least 1 of the 119 splicing-associated factors recently proposed to be most relevant to tumorigenesis[52] (Fig. 7f, Supplementary Data 5). While few of these mutations have been validated to affect splicing, the most common single mutation was at the known "hotspot" *SF3B1* K666T, found in three patients. Variant allele frequencies for these expected heterozygous mutations were 42%, 35%, and 22%, suggestive of a prominent subclonal fraction of the tumor cell population. Among well-characterized genes, mutations were found in *SF3B1* ($n = 10$ patients, including K666T), *SRSF2* ($n = 2$), *U2AF1* ($n = 4$), and *ZRSR2* ($n = 1$). Unfortunately, we were unable to obtain rare primary patient samples containing mutations in these genes, and no myeloma cell lines are known to carry hotspot mutations in these well-characterized splicing factors (www.keatslab.org). While our data suggest that spliceosome inhibition should be considered a therapeutic option for MM patients of any genotype, recent work in other malignancies[41,42] supports the potential for additional benefit in the subset of patients carrying pathogenic splicing factor mutations.

## Discussion

Our results demonstrate that PI therapy in myeloma leads to both specific alterations in splice site usage and broad-scale interference with spliceosome function. This observation, initially generated through unbiased phosphoproteomics, led us to explore the spliceosome itself as a MM vulnerability. Our preclinical evaluation and analysis of functional genomics and exome sequencing data further reinforced the spliceosome as a therapeutic target in MM.

These results raise a number of intriguing questions. From a mechanistic perspective, prior work examining SR phosphorylation after perturbation by western blotting did not consistently show a broad hyperphosphorylation signature[40,53,54]. Our results therefore illustrate the utility of unbiased phosphoproteomics to elucidate cancer drug response. Recent work also suggests that additional kinases beyond the well-characterized SRPKs and CLKs may be involved in SR phosphorylation[55,56]. The mechanism that leads to coordinated, upregulated phosphorylation across multiple splicing factors in the context of cancer therapy, and the specific role of caspases in this process, will be an important topic for future investigation.

We also found a correlation between Cfz-induced stress and both SR factor phosphorylation and the degree of alternative exon selection. Why these effects specifically lead to alternative splicing of protein homeostasis genes remains to be investigated. In contrast, we found IR to be largely independent of the degree of Cfz-induced stress. This result suggests that the IR phenotype is mediated by a different mechanism, uncoupled from SR factor phosphorylation. In addition, both our RNA-seq analyses and combination therapy experiments suggest that the irreversible PI, Cfz, carries different effects on splicing than the reversible PI, bortezomib. It is known these two agents are not identical, as evidenced by their broadly differing toxicity profiles, resistance

mechanisms, and clinical responses[57–59]. Further investigation is necessary to fully dissect why these two PIs carry differential effects on IR.

In attempting to model the relationship between SR phosphorylation and splicing, we recognize that our phosphomimetic construct does not fully recapitulate the complex phosphorylation biology of SRSF1 within cells[28] nor does it fully match the level of expression of endogenous SRSF1. Causal links have been noted between SRSF1 phosphorylation and splicing in single transcript, in vitro systems[35,60,61], but isolating global effects of SR phosphorylation on splicing within cells have remained elusive. Despite these limitations, however, our studies provide a landscape of the SRSF1 cytosolic and nuclear interactome, which may inform future studies of SR protein biology.

Here we propose that the loss-of-fitness modality of drug-induced IR constitutes a previously unexplored mechanism of action of Cfz. We further performed a preclinical evaluation of splicing inhibition in myeloma using E7107, finding potent anti-myeloma effects in vitro, in vivo, and ex vivo vs. primary patient samples. From a therapeutic perspective, one of the major questions is the potential toxicity of targeting the catalytic spliceosome. However, our analysis of genetic dependencies and our ex vivo treatment data clearly demonstrates the potential to target core spliceosome subunits in MM while largely sparing normal cells. In fact, based on this analysis the spliceosome appears to be an even more promising target than the clinically validated approach of targeting the proteasome. Furthermore, presumed efficacious doses (based on measured blood concentrations in the nM range) of E7107 were largely well tolerated in a Phase I clinical trial[62]. While this molecule is no longer in clinical development, it is thought that E7107 visual toxicity is molecule specific and is not a function of targeting the spliceosome in general[15]. Our genomic analysis suggests that mutations in splicing factors are found in a substantial fraction of MM patients. Newer generations of splicing inhibitors are currently in clinical trials for other hematologic malignancies[42] (NCT02841540) and may be of particular benefit for myeloma patients. Our results support clinical investigation of these compounds in MM, either alone or to enhance PI efficacy as combination therapy.

## Methods

**Cell culture**. All cell lines were grown in suspension at 37 °C, 5% $CO_2$ in complete media: RPMI 1640 medium (Gibco, 22400105, UCSF CCFAE002), supplemented with 10% fetal bovine serum (FBS; Atlanta Biologicals, S11150) for proteomics experiments or Benchmark FBS (Gemini Bio-products, 100–106) for drug viability experiments, and 1% penicillin–streptomycin (UCSF, CCFGK003). INA6 cell media was supplemented with 90 ng mL$^{-1}$ recombinant human IL-6 (ProSpec Bio, CYT-213).

**Drug cytotoxicity assay**. For dose–response cell toxicity assays, 1 E+3 myeloma cells were seeded per well in 384-well plates (Corning) using the Multidrop Combi (Thermo Fisher) and incubated for 24 h. In monotherapy cytotoxicity assays, cells were treated with drug or DMSO and incubated for 48 h, while cells were further incubated with E7107 (H3) for an additional 24 h in E7107 dual therapy combination assays. Cfz (Selleckchem, S2853), melphalan (Sigma, S2853), Bortezomib (Selleckchem, S1013), Venetoclax (LC Labs, V-3579), Lenalidomide (Sigma, CDS022536), E7107 (H3 Biomedicine, CAS:630100–90–2), and KH-CB19 (sc-362756) were solubilized in DMSO at 10 mM.

All cell viability was determined with Cell-Titer Glo reagent (Promega, G7573) using a Glomax Explorer (Promega) luminescence plate reader. For the drug titration cytotoxicity assays, measurements were performed in quadruplicate, while measurements were performed in triplicate in all other assays, and viabilities are reported as mean ($+/-$S.D.) ratio normalized to DMSO-treated controls or measurements at 0 h. For ZIP synergy calculations, normalized viability data were submitted to SynergyFinder web application[44].

**Drug dosing for proteomics and RNA-seq experiments**. Proteomic/phospho-proteomic/RNA-seq experiments were performed at a cell density of 1 E+6 cells mL$^{-1}$. For timecourse studies, ~20 E+6 cells were grown in complete media for

each timepoint (0, 8, 16, and 24 h), whereas for single-timepoint experiments, 15–20 E+6 cells in light SILAC media were treated with drug compound and cells in heavy SILAC media (L-Lysine-$^{13}C_6$,$^{15}N_2$, L-Arginine-$^{13}C_6$,$^{15}N_4$ (Cambridge Isotope, CNLM-291-H-1, CNLM-539-H-1) were treated with DMSO for 24 h. One-to-3 E+6 cells were set aside for RNA-seq. Cells were washed in phosphate-buffered saline (PBS) and cell pellets were frozen in liquid nitrogen (LN$_2$) and stored at −80 °C. One biological replicate for the timecourse experiment, two biological replicates for each single-timepoint condition (with a third only for RNA-seq), and three biological replicates for all AP-MS were gathered and analyzed. Single-timepoint experiments for MM.1S treated with 5 nM bortezomib or 50 μM lenalidomide or combination treatment of 15 nM Cfz and 50 μM TG003 (Selleckchem, S7320) or 15 nM Cfz and 50 μM z-VAD-FMK (Ubiquitin-Proteasome Biotechnologies, F7111) were carried out in 3 biological replicates.

**Cloning and lentiviral transduction**. *SRSF1* and *mCherry* genes, along with 3× FLAG sequences, with and without NLS were cloned into pLV-416G second-generation lentiviral plasmid (UCSF HMTB) by Gibson Assembly. SRSF1 constructs were transfected into Lenti-X 293T (Takara Bioscience, 632180) packaging cells with Gag-Pol expressing pCMV-dR8.91 (Addgene, Plasmid#2221) and VSV-G envelope expressing pMD2.G (Addgene, Plasmid#12259) plasmids. Viral particles were harvested and concentrated with Lenti-X concentrator (Takara Bioscience, 631231), and viral titers were incubated with AMO-1 cells. Positively transduced cells were selected with selection drug, G418 (VWR, 970-3-058), for several passages, then by mCherry expression with Fluorescence Activated Cell Sorting (FACS, Sony SH800). Protocol details are found in Supplementary Methods and Supplementary Table 1.

**Phosphoproteomic peptide preparation**. Frozen pellets of ~15–20 E+6 cells were lysed in 8 M urea, 0.1 M Tris pH 8.0, 150 mM NaCl, and 1× HALT phosphatase/ protease inhibitor cocktail (Pierce, 78442) for timecourse experiments or 8 M Guanadine-Cl (Gdn, Chem Impex Intl., 00152–1 KG), 0.1 M Tris pH 8.5, 10 mM tris(2-carboxyethyl)phosphine (TCEP, Pierce, 20491), 40 mM 2-chloroacetamide (2-CAA, Sigma, 22790–250G-F), 1× HALT for SILAC samples and lysed with probe sonicator (BRANSONIC). In the case of single-timepoint SILAC samples, equal part light- and heavy-labeled lysate samples were combined (~2.5–3 mg total). Lysate was diluted with 0.1 M Tris pH 8.0 to a final concentration of 1.3 M Gdn or urea. Proteome was digested with 1:100 dilution of trypsin overnight for 22–24 h at room temperature. Peptides are extracted with SEP-PAK C18 cartridges (Waters, WAT020515). For single-timepoint SILAC samples, ~100 μg of eluted peptides were dried and analyzed separately by liquid chromatography tandem mass spectrometry (LC-MS/MS) as unenriched "global proteomics." Remainder of eluate was diluted 3–4-fold with water, lyophilized, then resuspended in 80% acetonitrile (ACN, Sigma, 34998), 0.1% trifluoroacetic acid (Sigma, T6508–10AMP) and enriched on FeCl$_3$ charged NTA-agarose beads (VWR, 220006–720) sitting atop a C18 matrix in a stage-tip platform (Nest). Eluted phosphopeptides were dried and stored at −80 °C.

**Affinity purification**. For each replicate, frozen cell pellets were gently lysed on ice with 200 μL hypotonic lysis buffer (20 mM Tris (pH 7.4@4 °C), 10 mM KCl, 0.1 mM EDTA (Fisher, BP120–500), 0.5% NP-40 alternative (EMD, 492016–100ML), 1 mM dithiothreitol (DTT; Gold Biotech, DTT50), 1 mM phenylmethanesulfo-nylfluoride (PMSF; RPI, P20270–1.0), 1× HALT protease/phosphatase inhibitor cocktail (Pierce, 78442), 300 mM Sucrose, 0.03 U mL$^{-1}$ aprotinin (RPI, A20550–0.001)), underwent 3 freeze–thaw cycles, and clarified with 5 passes through an 18-gauge syringe needle. Lysate was centrifuged at 5000 rcf, 4 °C for 10 min, and supernatant was reserved as cytoplasmic fraction, while nuclear fraction was washed and resuspended in 60 μL of 20 mM HEPES (pH 7.9), 420 mM NaCl, 25% glycerol, 1 mM EDTA, 1 mM DTT, 1 mM PMSF, 0.03 U mL$^{-1}$ aprotinin, 1× protease/phosphatase inhibitor cocktail (HALT), 25 U Benzonase mL$^{-1}$ and clarified with 10 passes through 18-gauge syringe needle. Both fractions were adjusted to 50 mM Tris pH 7.4, 150 mM NaCl, and 1 mM EDTA (binding buffer) and combined with M2 anti-FLAG magnetic beads (Sigma, M8823). Bound lysate was washed with binding buffer + 0.05% NP-40, then binding buffer, then twice with 20 mM Tris, pH 8.0, and 2 mM CaCl$_2$. Proteins are denatured and cystines are reduced and alkylated with 6 M Gdn, 40 mM 2-CAA, 5 mM TCEP, 100 mM Tris pH 8.0, then trypsinized on-bead with ~0.75 μg trypsin per sample, ~20 h at 37 °C, and peptides were desalted with homemade C18 stage-tips and dried and stored at −80 °C.

**Liquid chromatography tandem MS/MS**. Approximately 1 μg peptides were analyzed for each sample by "shotgun" LC-MS/MS on a Dionex Ultimate 3000 RSLCnano with 15 cm Acclaim PEPMAP C18 (Thermo, 164534) reverse phase column and Thermo Q-Exactive plus mass spectrometer. Samples were analyzed with either a 3 h 15 min non-linear gradient or a 1 h 23 min linear gradient from 2.4% ACN (Sigma, 34998-4 L), 0.1% FA to 32% ACN. Experiment-specific LC-MS/MS settings are listed in Supplementary Information.

**Proteomic data analysis and quantification**. Initial timecourse unlabeled phosphoproteomics data were processed together on Maxquant v1.5.1.2[63] and searched

against the human proteome (Uniprot downloaded 3 Dec 2014, with 89,706 entries). All AP-MS samples were processed together with similar settings. All SILAC samples (phosphopeptides and unenriched peptides) were processed together with similar settings. SILAC quantification for global proteomics at the protein level requires one minimum razor or unique peptide. One-sample two-sided $t$ test was applied to the $\log_2$-transform of the normalized SILAC-labeled peptide ratios (heavy:light) for single-timepoint analysis, while for AP-MS data, two-sample $t$ test was applied to the $\log_2$-transform of the median-normalized MaxQuant label-free quantification values of the protein groups. The number of total entries (phosphosites, protein groups, significance is $p < 0.05$, |$t$ test difference| $\geq 1$), along with correlation statistics between replicates, are summarized in Supplementary Data 1 and shown in Supplementary Fig. 3b. See Supplementary Information for specific search and analysis settings.

**RNA-seq library preparation**. RNA was extracted from frozen cell pellets with the RNeasy Mini-prep Kit (Qiagen, 74104). For timecourse experiments, cDNA library of expression transcripts was carried out with the RNA Hyper Prep Kit with RiboErase (Kapa, KK8560) to enhance transcript reads above ribosomal reads, while single-timepoint experiments assessing splicing required mRNA enrichment with the magnetic mRNA Isolation Kit poly-dT beads (NEB), then the RNA Hyper Prep Kit (Kapa, KK8540) for cDNA construction of 200–400-bp library with Illumina platform TruSeq indexed adaptors (Supplementary Data 2). RNA and DNA quantified at all steps by Nanodrop (Thermo Scientific) and cDNA library size and quality were evaluated on a Bioanalyzer 2100 (Agilent) with the High Sensitivity DNA Kit (Agilent, 5067-4626), before being submitted for next-generation sequencing on a HiSeq4000 (Illumina) at the UCSF Center for Advanced Technologies core facility.

**JuncBASE alternative splicing analysis**. Alternative splicing events were identified and quantified with JuncBASE v1.2-beta using default parameters[22]. Intron-exon junction database was created from hg19 annotations. A $t$ test was used to compare number of inclusion and exclusion reads, and $p$ values were adjusted with Benjamini–Hochberg correction. For ΔPSI histograms in Figs. 3b, c, 4c, d, and 5a and Supplementary Figs. 4a, b and 7a–c, JuncBASE output included a subset of alternative splice events with median PSI = 0.00 in both conditions or median PSI = 100.00 in both conditions, resulting in ΔPSI = 0.00. These events were manually removed for downstream analyses. Histograms and splicing statistics were determined with statistical computing program R (v3.5.1) and a summary is listed in Supplementary Data 3.

**GO enrichment analysis**. GO enrichment analysis of upregulated phosphosites and enriched SRSF1 interactors was performed in STRING (v10.5, https://string-db.org/)[64], searching against a background of all quantified protein entries. Enrichment analysis of all significantly alternative spliced genes (raw $p < 0.05$) was performed using web-based enrichment analysis tool, Enrichr (v2.1, http://amp.pharm.mssm.edu/Enrichr/)[65]. Reported combined score is calculated by multiplying the natural log transform of the $p$ value with the Fisher's exact test of expected rank deviation (Z-score). Functional GO analysis is limited to biological processes and compiled in Supplementary Data 4.

**Xenograft mouse model and in vivo luminescence imaging**. 1 E+6 MM.1S-luc cells, stably expressing luciferase, were transplanted via tail vein injection into 12 NOD.Cg-*Prkdc*$^{scid}$ *Il2rg*$^{tm1Wjl}$/SzJ (NSG) mice from The Jackson Laboratory (cat# 005557). All the mice were female, 6–8-week-old at the start of studies, and typically weigh 20–25 g. NSG mice were handled with aseptic techniques and housed in pathogen-free environments at the UCSF Laboratory Animal Research Center barrier facility. Female mice were housed with five mice per cage. All mouse studies were performed according to UCSF Institutional Animal Care and Use Committee-approved protocols. Tumor burden was assessed through weekly bioluminescent imaging in the UCSF preclinical therapeutic core on a Xenogen In Vivo Imaging System, beginning 13 days after implantation, which is the same day as treatment initiation. Tumor-implanted humanized mice were randomized and sorted into control arm and treatment arm, six mice per arm. Mice were treated for 2 weeks (5 days on, 2 days off) with vehicle or 3 mg kg$^{-1}$ E7107, formulated in vehicle (10% ethanol, 5% Tween-80, QS with saline) and administered by continuous subcutaneous infusion. Mice were kept and observed until survival endpoint; final timepoint was 54 days after MM.1S transplant. Acquired luciferase intensities were quantified with the Living Image Software (PerkinElmer) in units of radiance (photons s$^{-1}$ cm$^{-2}$ sr$^{-1}$). Kaplan–Meier survival curves along with log-ranked test to determine significance were calculated in the GraphPad Prism 6 software.

**Patient sample analysis**. Fresh de-identified primary MM patient BM samples were obtained from the UCSF Hematologic Malignancies Tissue Bank in accordance with the UCSF Committee on Human Research-approved protocols and the Declaration of Helsinki. BM mononuclear cells were isolated by density gradient centrifugation with Histopaque-1077 (Sigma Aldrich), and washed with 10 mL D-PBS 3 times. Mononuclear cells were resuspended in a small volume (~1.5 mL) of media (RPMI1640, 10% FBS, 1% penicillin/streptomycin, 2 mM glutamine) and

incubated at 37 °C, 5% $CO_2$ for 15 min. Isolated mononuclear cells from MM patient BM were adjusted to 2 E+5 cells per well in a 96-well plate. Cells were stimulated with 50 ng mL$^{-1}$ recombinant human IL-6 (ProSpec) for 17 h before treatment with E7107 or DMSO for 24 h. Cells were then stained with 10 μL Alexa-Fluor 647 mouse anti-human CD138 antibody (BD Pharmingen, cat# 562097; RRID:AB_10895974) or Alexa-Fluor 647 IgG κ isotype (BD Pharmingen, cat# 557714; RRID:AB_396823) control and 2 μL SyTOX Green (Thermo, S34860) per 1 mL FACS buffer (D-PBS, 5% FBS). Resuspended cells are characterized with a CytoFLEX fluorescence cytometer (BD).

**Statistical analyses**. All data are presented as mean +/− standard deviation, unless otherwise stated. Phosphoproteomics and global proteomics are shown with two biological replicates, while RNA-seq and AP-MS were conducted in three biological replicates. All other statistics, unless otherwise stated, are presented as technical replicates. Statistical significance in proteomics comparisons was determined by Student's two-tailed $t$ test: one-sample $t$ test with null hypothesis that $\log_2$-transform of the normalized SILAC ratio is equal to 0 or two-sample $t$ test with null hypothesis that the difference in $\log_2$-transform of the intensities is equal to 0. A $p < 0.05$ is considered statistically significant. For all Kaplan–Meier survival analysis, log-ranked test was used to determine statistical significance.

**Reporting summary**. Further information on research design is available in the Nature Research Reporting Summary linked to this article.

## Data availability

The mass spectrometric proteomics data and MaxQuant analysis results have been deposited to the ProteomeXchange Consortium via the PRIDE repository with the dataset identifier PXD012172. Raw RNA-seq data, processed analysis files, and JuncBASE results may be downloaded from the Gene Expression Omnibus, GEO with the accession number: GSE124510. All relevant data are available from the authors. The source data underlying Fig. 6e and Supplementary Figs. 3c–g and 4e are provided as a Source Data file.

## Code availability

Current and previous releases of JuncBASE can be found on github: https://github.com/anbrooks/juncBASE.

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

## Acknowledgements

We thank Dr. Silvia Buonamici at H3 Biomedicine for providing E7107 and insightful discussions and Jacob Runyan for assisting in quality control of sequencing data. We thank Dr. Renate Burger and Dr. Christoph Driessen for providing INA-6 and AMO-1 parental and bortezomib-resistant cell lines, respectively. We also thank the laboratory of James Wells for use of Zeiss Z1 Observer microscope, Center for Advanced Technology at UCSF for HiSeq sequencing, Dr. Jane Gordon and the Laboratory for Cell Analysis at Helen Diller Family Comprehensive Cancer Center for use and assistance of Sony SH800, and Dr. Danielle Swaney for discussion of AP-MS. This work was supported by NIH/NCI P30CA083103 (Cancer Center Support Grant, supporting UCSF Preclinical Therapeutic Core facility managed by B.C.H.), NIH/NHGRI T32HG008345 (to A.M.T.), the Damon Runyon Cancer Research Foundation Dale Frey Breakthrough Award (DFS 14-15), NIH/NCI K08CA184116, NIH/NIGMS DP2OD022552, the UCSF Stephen and Nancy Grand Multiple Myeloma Translational Initiative (to A.P.W.), and NIH/NCI R01CA226851 (to A.N.B. and A.P.W.).

## Author contributions

H.H.H., I.F., C.L., P.B., P.C., M.C.M., C.F.T., M.D.M., and A.P.W. performed experiments and analyzed experimental data. A.M.T., Y.-H.L., and A.N.B. analyzed transcriptomic and genomic data. J.M., P.J.P., and B.H. performed in vivo studies. T.G.M., J.L.W., S.W.W., and N.S. obtained consent from patients and also primary specimens. H.H.H. and A.P.W. wrote the manuscript with input from all authors.

## Competing interests

The authors declare no competing interests.
