## [Peer Review File · Nature Communications]

Reviewers' comments:

Reviewer #1 (Remarks to the Author):

This is an interesting manuscript which utilizes phosphoproteomics, total proteomics, RNA sequencing and a number of other assays to identify that proteasome inhibitor (PI) treatment in multiple myeloma perturbs RNA splicing through effects on phosphorylation of RNA splicing factors. There are lot of high quality data presented. Nonetheless, it would be very helpful if the authors could address/clarify a number of points as follows:

- How does PI alter phosphorylation of RNA splicing factors? This question doesn't seem to be addressed in the manuscript.
- Can the authors delineate which splicing factor/modification of splicing factor is responsible for the splicing changes seen with PI use? It does not seem likely that cleavage of SF3B1 or U2AF1 is responsible for the effects of PI on RNA splicing since apparently there were no major changes in cassette exon usage with PI (where SF3B1 loss/inhibition would result in widespread exon skipping as seen here with E7107 and in prior papers with U2AF1 genetic loss (see PUBMED ID 25267526).
- Other than the data in Figure S4g there are very few illustrations of individual splicing events which are impacted by PI treatment in myeloma cells. It would be very helpful to demonstrate time/dose-dependent effects of PI on splicing of some high confidence aberrant splicing events associated with PI.
- It is hard to understand the role of SF3B1 expression in response to spliceosome inhibition in Figure 6 given that mRNA and protein expression for SF3B1 do not seem to be correlated. Are there differences in splicing across the cell lines which are "less sensitive" versus "more sensitive" to SF3B1 inhibition? Do these differences hold up in in vivo treatment (as was done for MM.1S cells in Figure 7a-c)?

Reviewer #2 (Remarks to the Author):

In this manuscript, Huang and colleagues used approaches based on phosphoproteomics methods to investigate signaling responses triggered by the irreversible proteasome inhibitor carfilzomib. With this approach, the authors discovered that these treatments induce the phosphorylation of proteins present in spliceosome complexes. GO enrichment analysis confirmed that RNA splicing was the most enriched ontology after treatment with the PI. After validating that carfilzomib disrupts the process of splicing, the authors show that treatment of cells with a small molecule spliceosome inhibitor synergizes with PI in reducing the viability of MM cell cultures and it is effective in reducing tumor growth in in vivo models and in patient samples. The study is interesting, well written and shows an elegant approach to identify new drug targets in cancer. There however are some minor issues that the authors should address before publication.

1. The omics data are deposited in repositories and this is commendable. However, phosphopeptides in supplementary tables (excel files) should also contain some information on the quality of the identification with well annotated with columns indicating gene names or Uniprot IDs, scores of identification, position of modified residue within the protein, etc. Otherwise these tables will not be useful to the interested reader.
2. The authors should change the term "Kinase Set enrichment analysis" (lane 197 in page 7, and 179 in suppl data) to "Kinase Substrate Enrichment Analysis".
3. A finding of the study is that CFZ treatment induces apoptosis. The authors could also mention that this is consistent with their KSEA analysis showing an increase in of DNA-PK and ATM kinase activities, as these kinases are activated during apoptosis.
4. On page 3, lane 55, the word "incredible" does not seem to be used appropriately in this context.

Reviewer #3 (Remarks to the Author):

In this manuscript Haung et al. performed a comprehensive study to better understand the PI mechanism of actions, sensitivity and resistance using phosphoproteomics, functional genomics and transcriptomics. Manuscript was very well written and points were clearly and fairly made. However, although experiments were well designed, figures were not capable of depicting the points that described in the main text.

For instance:

Using the same heatmap scale for both phosphorylation and gene expression levels in Fig. 1b , made it very difficult to appreciate altered gene expression patterns at different time-points for most of differentially expressed genes such as RSP6 and SRSF proteins.

Cell viability curves and ZIP-synergy heatmaps in Fig. 6 did not display any benefits with Cfz-E7107 vs Melphalan-E7107. Better plotting the results might help to reinforce the authors conclusion, in particular to exclude the fact that most of anti-MM activity caused by E7107.

Using phosphoproteomics to study drug-mediated responses was a novel approach in multiple myeloma, however authors did not demonstrate the mechanism underlying altered activity of splicing factors such as SRSF1.

Altered expression and activity of splicing-associated factors have been described in many cancer types including MM and spliceosome inhibitors are in clinical trials for solid tumors.

This study proposed splicing interference as a new mechanism of action for PIs, but did not provide good evidence supporting that scenario. In addition, Cfz-E7107 combo-therapy failed to describe benefits to Melphalan-E7107 treatment, further revealing that proposing a novel regulatory function for PIs in alternative splicing and intron retention requires further investigation.

Arun P. Wiita, MD, PhD
Associate Professor in
Residence

Assistant Director,
Clinical Cytogenetics
Laboratory

Director, Stephen and
Nancy Grand Multiple
Myeloma Translational
Initiative Laboratory

Department of
Laboratory Medicine
185 Berry St.
Suite 290, Room 2410
San Francisco, CA
94107

tel: (415) 514-6238
email:
arun.wiita@ucsf.edu

Reviewer 1:

This is an interesting manuscript which utilizes phosphoproteomics, total proteomics, RNA sequencing and a number of other assays to identify that proteasome inhibitor (PI) treatment in multiple myeloma perturbs RNA splicing through effects on phosphorylation of RNA splicing factors. There are lot of high quality data presented. Nonetheless, it would be very helpful if the authors could address/clarify a number of points as follows:

1. How does PI alter phosphorylation of RNA splicing factors? This question doesn't seem to be addressed in the manuscript.

Response: This question is certainly of significant interest to us as well. To address the possible mechanisms of splicing factor phosphorylation, we pursued several additional phosphoproteomic experiments, presented in the new Supplementary Fig. S6. First, to evaluate whether the splicing factor phosphorylation phenomenon was unique to carfilzomib, we treated MM.1S cells with the reversible PI bortezomib, also a first-line therapy in myeloma. Notably, this dose induced ~85% cell death (revised Supplementary Fig. S3a), similar to the conditions for our initial carfilzomib experiments (Fig. 1b and 2A). Under these conditions, we again noted prominent phosphorylation of SR splicing factors (orange dots in Supplementary Fig. S6a), similar to our prior findings with carfilzomib. In contrast, we treated MM.1S cells with 50 uM lenalidomide, a thalidomide analog and another first-line myeloma therapy. MM.1S is known to be largely resistant to lenalidomide even at this very high dose, and indeed we found essentially no change in phosphorylation. This finding is consistent with a mechanism where SR phosphorylation is related to the degree of cell death, as our prior dose-dependent carfilzomib studies suggested (Fig. 2). Furthermore, as SR splicing factors are known to be phosphorylated by the CLK kinase family, we simultaneously treated MM.1S cells with both carfilzomib (cfz) and TG003, a potent inhibitor of CLK1 and 4 (Muraki et al, J Biol Chem (2004) 279:24246) versus cfz alone. We performed this experiment for 24 hr to allow time for the cfz response to occur (see Fig. 1B). We expected to see blockade of SR phosphorylation under this combination treatment. Surprisingly, we actually saw *increased* SR phosphorylation under the cfz + TG003 condition. We interpret this result as most likely indicative of rebound phosphorylation of substrates under sustained kinase inhibition, a well-known phenomenon (we have noted it most recently in phosphoproteomics of JAK inhibition in myeloma cells at 24 h vs. 1.5 hr of treatment (Lam et al, *Haematologica* (2018) 103:1218)). Therefore, as we indeed saw a change in SR phosphorylation status with TG003 treatment, this result is suggestive the CLK family kinases are involved in the cfz response. However, we cannot definitively rule out alternate kinase involvement. Our final experiment, though, proved more revealing. We compared cfz alone to cfz + 50 uM z-VAD-fmk, the pan-caspase inhibitor (Supplementary Fig. S6d). Remarkably, despite the

Arun P. Wiita, MD, PhD
Associate Professor in
Residence

Assistant Director,
Clinical Cytogenetics
Laboratory

Director, Stephen and
Nancy Grand Multiple
Myeloma Translational
Initiative Laboratory

Department of
Laboratory Medicine
185 Berry St.
Suite 290, Room 2410
San Francisco, CA
94107

tel: (415) 514-6238
email:
arun.wiita@ucsf.edu

fact that the majority of phosphorylation events detected showed moderate (less than 4-fold) upregulation, almost all the detected SR domain phosphorylation events showed *downregulation*. Therefore, our results suggest that, mechanistically, caspase activation is a key step in leading to SR domain phosphorylation, most likely by the CLK family kinases. This model is also consistent with our prior results, where the degree of SR domain phosphorylation correlated with the degree of cell death (Fig. 2a,c). Further investigation, beyond the scope of this work, will focus on attempting to identify the specific links between caspase activity and activation of CLKs.

Related to these findings, we have added additional description to the main text on p. 14-15.

2. Can the authors delineate which splicing factor/modification of splicing factor is responsible for the splicing changes seen with PI use? It does not seem likely that cleavage of SF3B1 or U2AF1 is responsible for the effects of PI on RNA splicing since apparently there were no major changes in cassette exon usage with PI (where SF3B1 loss/inhibition would result in widespread exon skipping as seen here with E7107 and in prior papers with U2AF1 genetic loss (see PUBMED ID 25267526)).

Response: We certainly agree this is an interesting biological question to address. Toward one possible mechanism, our results with low doses of carfilzomib, inducing minimal cell death, indeed still led to significant intron retention (Fig. 5a). However, we do note that intron retention found under carfilzomib treatment is still dramatically less than that seen with potent blockade by E7107 (compare Fig. 5a and Fig. 6a). Therefore, we cannot rule out that a scenario where even in the setting of ~10% cell death, a low level of caspase cleavage of SF3B1 and U2AF1 (i.e. not resolvable by Western blot) is enough to lead to the degree of intron retention as we find here. This model could still be consistent with the prior SF3B1 inhibition and U2AF1 genetic depletion studies cited by the reviewer, where much greater loss-of-function leads to much greater intron retention. However, we also agree it is unlikely to be the case, particularly given the contrast we see to melphalan with similar levels of cell death but much less intron retention (p. 10 of the manuscript). Furthermore, high levels of cell death after 5 nM bortezomib (Supplementary Fig. S2A) still did not lead to significant intron retention (Supplementary Fig. S7A). Therefore, it appears that other mechanisms are much more likely to be in play.

To find alternate mechanistic explanations, we further examined the literature describing the effects of heat shock on splicing. As we note in the manuscript, heat shock is a prominent response to proteasome inhibition and we hypothesized it may play a role in the splicing dynamics found here. As described in our introduction, heat shock was first noted by Susan Linquist and colleagues to affect splicing of well-described, canonical transcripts in *Drosophila* models in 1986. Since then, however, surprisingly little headway has been made in elucidating what actually causes these splicing changes. Essentially the only paper we could identify proposing a clear mechanistic explanation came from Jim Manley's lab at Columbia, where they suggested that hypophosphorylation of SRSF10/SRp38 was the

Arun P. Wiita, MD, PhD
Associate Professor in
Residence

Assistant Director,
Clinical Cytogenetics
Laboratory

Director, Stephen and
Nancy Grand Multiple
Myeloma Translational
Initiative Laboratory

Department of
Laboratory Medicine
185 Berry St.
Suite 290, Room 2410
San Francisco, CA
94107

tel: (415) 514-6238
email:
arun.wiita@ucsf.edu

critical mediator of heat shock-induced splicing alterations (Shin et al, Nature (2004) 427:553). To test if this mechanism may be active here, we performed Western blotting for SRSF10 after carfilzomib treatment for 18 h at 5 and 10 nM doses (revised Supp. Fig. S3g). While HSP27 blotting confirmed induction of the heat shock response, unlike in the Manley paper, we saw no evidence of any gel shift in SRSF10 to indicate hypophosphorylation (notably, re-analysis of our phosphoproteomic datasets also showed no evidence of SRSF10 phosphopeptide downregulation (not shown)). We did note significant loss of full-length SRSF10 at the higher dose, consistent with this protein being a caspase substrate. Therefore, an alternate hypothesis is that partial loss of SRSF10, even at low levels of caspase cleavage, could also be causative of the intron retention we see here.

We further performed additional experiments using our mCherry-labeled SRSF1 constructs. We hypothesized that altered localization of SR factors, particularly nuclear-to-cytosolic shuttling, could possibly lead to some of the alterations in donor and acceptor patterns found in Fig. 5b. As stated in the manuscript, we continued to focus on SRSF1 as it is the canonical and most-studied member of the SR family. We therefore performed confocal microscopy on AMO1 cells treated with DMSO or cfz, using a WT SRSF1-FLAG-mCherry construct (note that this construct is similar to those used for AP-MS experiments in Fig. 4, but the construct here did not have a nuclear localization signal). Even without the NLS, we found strong localization of SRSF1 to the nucleus. Under cfz doses leading to ~10% apoptosis (measured by morphology of pyknotic nuclei), we found no evidence of any shuttling of SRSF1 out of the nucleus to the cytosol (new Supplementary Fig. S8). A FLAG-mCherry alone construct (also with no NLS) served as a control for primarily cytosolic staining. In this data, we did also note a possible mild morphologic phenotype, whereby cfz-treated cells showed a trend toward a distribution of SRSF1 from diffusely throughout the nucleus more to the nuclear periphery (new Supp. Fig. S8f). However, it is unclear whether this putative change in SRSF1 localization is enough to drive the splicing changes noted here.

Taken together, we believe these additional studies underscore that it is most likely there are several mechanisms working in parallel to lead to the altered splicing patterns we find after cfz treatment. As emphasized above, a more complete delineation of this mechanism appears beyond the scope of this work and will be tackled with extensive, independent follow-up.

To describe the results here, we have extensively altered the main text on pages 14-15.

3. Other than the data in Figure S4g there are very few illustrations of individual splicing events which are impacted by PI treatment in myeloma cells. It would be very helpful to demonstrate time/dose-dependent effects of PI on splicing of some high confidence aberrant splicing events associated with PI.

Response: In response to this query, we reasoned that examining the effects of splicing

Arun P. Wiita, MD, PhD
Associate Professor in
Residence

Assistant Director,
Clinical Cytogenetics
Laboratory

Director, Stephen and
Nancy Grand Multiple
Myeloma Translational
Initiative Laboratory

Department of
Laboratory Medicine
185 Berry St.
Suite 290, Room 2410
San Francisco, CA
94107

tel: (415) 514-6238
email:
arun.wiita@ucsf.edu

after treatment with the PI bortezomib (btz) may be a preferred approach to define generalized PI-associated splicing events. In RNA-seq of MM.1S samples treated with 5 nM btz for 24 h, which led to ~85% cell death, we surprisingly did not find any notable global intron retention response. However, we did see modest alterations in cassette exon and alternate first exon usage (new Supplementary Fig. S7a), indicating some changes in splicing post-therapy. However, toward defining aberrant splicing effects associated with PI, we did note overlap with several splicing events also noted after cfz treatment, including the proteasome subunits *PSMC5*, *PSMD13*, and *PSMB7*. These are illustrated in revised Supplementary Fig. 7d. Taken together, our results suggest that PI treatment converges on alterations of exon useage/splice donor/acceptor in the ubiquitin-proteasome pathway.

We do also note that, surprisingly, heat shock for 4 hr at 42 C, where we saw prominent HSP27 induction, we found minimal global changes in splicing. This result appears somewhat contradictory with the findings of Shalgi et al. (Cell Rep (2014) 7:1362), though even in that paper the authors note that mild heat shock led to significantly less intron retention than strong heat shock leading to widespread cell death. Therefore, it is possible that we would have found intron retention if we performed a longer heat shock, where our pilot studies at 8 h found widespread cell death (not shown). However, we note we chose these conditions to be most equivalent to the low-dose carfilzomib results in Fig. 5a.

Our results in response to question 2, the lack of notable splicing alteration under mild heat shock, as well as the lack of intron retention after bortezomib treatment, leads us to revise our tentative model in Fig. 5e. We are now less certain that the heat shock response is the primary driver of intron retention we see after carfilzomib. However, given that btz treatment appears to decouple the splicing factor phosphorylation and alternative exon usage from intron retention, we are more confident that there are two parallel pathways at play, both ultimately converging to alter splicing in plasma cells. However, many details of this mechanism clearly remain to be elucidated.

In addition to revising the model in Fig. 5, we have further modified the text on p. 14-15 following statements to the text to reflect this new experimental data.

4. It is hard to understand the role of SF3B1 expression in response to spliceosome inhibition in Figure 6 given that mRNA and protein expression for SF3B1 do not seem to be correlated. Are there differences in splicing across the cell lines which are “less sensitive” versus “more sensitive” to SF3B1 inhibition?

Response: The differential sensitivity between cell lines is certainly intriguing to us as well, and the lack of mRNA-protein correlation was certainly disappointing in terms of identifying an obvious biomarker of response. The suggestion here is certainly an intriguing one to characterize additional differences between sensitive and insensitive lines. Due to resource constraints we did not proceed with paired-end RNA-seq for all cell lines in the panel in triplicate, as would be necessary for a truly rigorous analysis.

Arun P. Wiita, MD, PhD
Associate Professor in
Residence

Assistant Director,
Clinical Cytogenetics
Laboratory

Director, Stephen and
Nancy Grand Multiple
Myeloma Translational
Initiative Laboratory

Department of
Laboratory Medicine
185 Berry St.
Suite 290, Room 2410
San Francisco, CA
94107

tel: (415) 514-6238
email:
arun.wiita@ucsf.edu

However, as a more preliminary investigation we did compare JuncBASE findings between MM.1S (insensitive) and AMO1 (sensitive) cells, using data that we had already collected. Comparison between these two lines (new Supplementary Fig. 7c) unfortunately did not reveal any clear signatures of global splicing alterations to help dissect this phenotype.

5. Do these differences hold up in in vivo treatment (as was done for MM.1S cells in Figure 7a-c)?

Response: This is also a great question as to whether this differential sensitivity is only an *in vitro* phenomenon. We do note that only three of the cell lines profiled (MM.1S (insensitive), JJN-3 (insensitive), RPMI-8226 (sensitive) home to murine bone marrow when implanted intravenously into NSG mice. The other cell lines can only be used in subcutaneous xenograft implantation, a less physiologically-relevant approach that is not comparable to our disseminated MM.1S luciferase model from an *in vivo* drug distribution perspective. We therefore attempted to perform *in vivo* studies using a stably luciferase-labeled RPMI-8226 cell line we had available in the lab (Sherbenou et al, J. Clin. Invest (2016) 126:4640). However, we saw aberrant consistency in bone marrow implantation and spontaneous tumor regression of the cells after establishment of bioluminescent signal, which we had not seen before (not shown). This phenomenon was true using different vials of cell stocks and precluded drawing any conclusions from therapy administration. We are unclear as to why we saw this unusual phenotype but we may need to re-derive this luciferase-labeled RPMI-8226 cell line, starting from lentiviral transduction and selection, to complete this experiment. If absolutely critical for publication we could pursue this, but given that this appears to be a useful, but not essential, experiment, we would respectfully ask that we be able to forgo it.

Reviewer 2:

The study is interesting, well written and shows an elegant approach to identify new drug targets in cancer. There however are some minor issues that the authors should address before publication.

1. The omics data are deposited in repositories and this is commendable. However, phosphopeptides in supplementary tables (excel files) should also contain some information on the quality of the identification with well annotated with columns indicating gene names or Uniprot IDs, scores of identification, position of modified residue within the protein, etc. Otherwise these tables will not be useful to the interested reader.

Response: Thank you for pointing out this omission. We have included the Maxquant phosphosite search results for the timecourse phosphoproteomic data as a tab S1.1 in Supplementary dataset S1, and updated the timecourse dataset by parsing gene name and peptide sequence. The remaining datasets include Uniprot entries, gene names, and phosphosite positions; these lists were processed from Maxquant search and further filtered in Perseus, as mentioned in the supplemental methods. We have updated these methods to include the step that all phosphopeptides with localization scores below 0.75 were excluded from analysis. These analyzed results can be linked back to tab S1.1 to find

Arun P. Wiita, MD, PhD
Associate Professor in
Residence

Assistant Director,
Clinical Cytogenetics
Laboratory

Director, Stephen and
Nancy Grand Multiple
Myeloma Translational
Initiative Laboratory

Department of
Laboratory Medicine
185 Berry St.
Suite 290, Room 2410
San Francisco, CA
94107

tel: (415) 514-6238
email:
arun.wiita@ucsf.edu

confidence scores for peptide identification and phosphosite localization. The Maxquant search files for all other proteomic data are included in the PRIDE repository (PXD012172) and can be downloaded (see data availability for details). To include all the search results for all additional experiments with the Supplementary dataset would greatly increase the file size; we believe it is reasonable that interested readers could access this processed data via the repository site for the other phosphoproteomic experiments.

2. *The authors should change the term “Kinase Set enrichment analysis” (lane 197 in page 7, and 179 in suppl data) to “Kinase Substrate Enrichment Analysis”.*

Response: Thank you for pointing out this error. We have now corrected it.

3. *A finding of the study is that CFZ treatment induces apoptosis. The authors could also mention that this is consistent with their KSEA analysis showing an increase in of DNA-PK and ATM kinase activities, as these kinases are activated during apoptosis.*

Response: This is an excellent biological point and we have modified the manuscript on p. 7 to include this observation.

4. *On page 3, lane 55, the word “incredible” does not seem to be used appropriately in this context.*

Response: We have corrected this to “extremely high”.

Reviewer 3:

In this manuscript Haung et al. performed a comprehensive study to better understand the PI mechanism of actions, sensitivity and resistance using phosphoproteomics, functional genomics and transcriptomics. Manuscript was very well written and points were clearly and fairly made. However, although experiments were well designed, figures were not capable of depicting the points that described in the main text.

For instance:

1. *Using the same heatmap scale for both phosphorylation and gene expression levels in Fig. 1b made it very difficult to appreciate altered gene expression patterns at different time-points for most of differentially expressed genes such as RSP6 and SRSF proteins.*

Response: We understand the point of the reviewer here, but we specifically chose the same scale bar for the phosphoproteomic and RNA-seq data to emphasize the differential fold-changes between these two measurements. We wanted to specifically point out that none of these gene expression changes would be detected as particularly prominent using any standard fold-change cutoffs, but that proteins encoded by these genes exhibit very prominent fold-changes at the phosphoproteomic level. We believe that this point would be lost if we used different scale bars for the RNAseq and phosphoproteomic data. If readers are interested in re-plotting or re-interpreting this data in any way, it is available in the Supplementary material. We would respectfully ask that we are able to leave this figure unaltered.

Arun P. Wiita, MD, PhD
Associate Professor in
Residence

Assistant Director,
Clinical Cytogenetics
Laboratory

Director, Stephen and
Nancy Grand Multiple
Myeloma Translational
Initiative Laboratory

Department of
Laboratory Medicine
185 Berry St.
Suite 290, Room 2410
San Francisco, CA
94107

tel: (415) 514-6238
email:
arun.wiita@ucsf.edu

2. Cell viability curves and ZIP-synergy heatmaps in Fig. 6 did not display any benefits with Cfz-E7107 vs Melphalan-E7107. Better plotting the results might help to reinforce the authors conclusion, in particular to exclude the fact that most of anti-MM activity caused by E7107.

Response: We agree with the reviewer's point that the ZIP synergy plots are somewhat confusing. In response to both this query and query #4 below, we have performed several additional combination therapeutic studies to specifically examine the effects of carfilzomib in combination with E7107. Furthermore, we have eliminated the synergy plot and instead shown a bar graph of the overall synergy score (revised Fig. 6 with bar graph, additional raw cytotoxicity curves in new Supplementary Fig. S10) to increase clarity for the reader.

In terms of the specificity of carfilzomib interaction with E7107, upon our further investigation we found several interesting results. First, we performed our combination assay in two different cell lines (AMO1 and MM.1S) and found equivalent strong synergy with E7107 and cfz in both cell lines (ZIP score ~6), which is essentially the same as we had found in the initial version of our manuscript. We notably saw no evidence of synergy of HS5 bone marrow stromal cells, suggesting this synergy may be, at least to some degree, plasma-cell specific. Furthermore, we performed the same assay with bortezomib and saw much weaker synergy (ZIP score ~2) in both MM.1S and AMO1 cell lines. This result actually appears very consistent with our splicing analysis described in response to Reviewer 1: as bortezomib does not appear to induce the same degree of intron retention as carfilzomib, it also does not appear to have the same degree of synergy with E7017. This is consistent with our hypothesis in the manuscript that broadly interfering with the splicing machinery (as noted by the phenotypic output of intron retention) in two different ways may be beneficial for driving plasma cell death. We do acknowledge, though, that it is unclear why bortezomib and carfilzomib appear to have these differential effects on intron retention, though it may relate to the irreversible nature of cfz. We do note that cfz has been clinically noted to drive deeper remissions than bortezomib; this IR phenotype may play a role in this improved response. As noted in the response to Reviewer 1, this finding will be part of follow up study beyond this work.

These additional studies also confirmed that synergy with E7107 + melphalan is much weaker than E7107 + carfilzomib. In fact, on this analysis melphalan actually appeared to be weakly antagonistic with E7107 (ZIP score ~ -2) as opposed to weakly synergistic in our prior results (ZIP score ~ +2) (see old and new figure comparison below). However, in both the (a) old and (b) new data, it is clear upon visual inspection that melphalan does not clearly shift the E7107 LC50. This finding suggests that the ZIP score may not be very accurate in this regime of +/-2, i.e. without either strong synergy or antagonism, where the result is essentially just additive. Overall, this reproduced result is again supportive of the hypothesis that E7107 synergy with melphalan is much less than that with carfilzomib.

Arun P. Wiita, MD, PhD
Associate Professor in
Residence

Assistant Director,
Clinical Cytogenetics
Laboratory

Director, Stephen and
Nancy Grand Multiple
Myeloma Translational
Initiative Laboratory

Department of
Laboratory Medicine
185 Berry St.
Suite 290, Room 2410
San Francisco, CA
94107

tel: (415) 514-6238
email:
arun.wiita@ucsf.edu

We further examined two additional combinations with E7107. First, we examined Lenalidomide, which we found to have no viability effect up to 50 μM in MM.1S cells. Combination with E7107 did not lead to any re-sensitization of MM.1S cells to Lenalidomide. While the ZIP score cannot be calculated when there is no viability effect at all from one of the drugs, again upon manual inspection it is clear there is no evidence of synergy (Supplementary Fig. 10). Finally, we also tested a combination of E7107 with the emerging myeloma therapy venetoclax (revised Fig. 6h). We were excited to see equivalent synergy between this drug + E7107 and the carfilzomib + E7107 combination, apparent both by the ZIP score and raw viability curves. Emerging evidence suggests that proteasome inhibitor resistance is largely mediated by effects on mitochondrial metabolism (Tsetsov et al., *Nat Chem Biol* (2019) 15:681; Besse et al, *Haematologica* (2019) 104:e415), and targeting mitochondria with venetoclax may be one way to overcome this resistance. This intriguing finding will also be the result of further study, beyond the scope of this work.

Overall, these results more firmly support our hypothesis regarding synergy between two different modes of intron retention induced by Cfz and E7107. Furthermore, the venetoclax-E7107 synergy is certainly a tantalizing finding. We believe that these additional experiments are sufficient to warrant further pre-clinical, and possibly clinical, investigation of the use of clinical-grade splicing inhibitors in myeloma.

As a result of these findings, in addition to the new addition to the Supplementary Figures as outlined, we have added new descriptive text on p. 18 of the manuscript.

3. Using phosphoproteomics to study drug-mediated responses was a novel approach in multiple myeloma, however authors did not demonstrate the mechanism underlying altered activity of splicing factors such as SRSF1. Altered expression and activity of splicing-associated factors have been described in many cancer types including MM and spliceosome inhibitors are in clinical trials for solid tumors.

Response: We agree that additional mechanistic insight into this system is warranted. We refer to our response to the mechanistic questions of Reviewer #1 above. Overall, our results suggest that the mechanism underlying this interaction is clearly very complex. Based on our additional extensive experiments performed as part of this revision, we believe that we have provided sufficient depth of study to justify this initial dissemination of results. However, we certainly agree that even further mechanistic investigation in a separate, follow-up study will be necessary to delve even deeper into what is “special”

Arun P. Wiita, MD, PhD
Associate Professor in
Residence

Assistant Director,
Clinical Cytogenetics
Laboratory

Director, Stephen and
Nancy Grand Multiple
Myeloma Translational
Initiative Laboratory

Department of
Laboratory Medicine
185 Berry St.
Suite 290, Room 2410
San Francisco, CA
94107

tel: (415) 514-6238
email:
arun.wiita@ucsf.edu

about splicing in myeloma. We do also note that clinical trials of E7107 in solid tumors have been abandoned due to unacceptable toxicities. Part of our goal with this paper is to motivate clinical studies of newer splicing inhibitors, such as H3B-8800, in multiple myeloma, which was not an initial indication for this compound. We would argue that these observations certainly go beyond what was already known in the field, particularly in terms of the apparently exquisite sensitivity of myeloma plasma cells to splicing inhibition compared to other tumor types.

4. This study proposed splicing interference as a new mechanism of action for PIs, but did not provide good evidence supporting that scenario. In addition, Cfz-E7107 combination therapy failed to describe benefits to Melphalan-E7107 treatment, further revealing that proposing a novel regulatory function for PIs in alternative splicing and intron retention requires further investigation.

Response: See response to query #2 above.

REVIEWERS' COMMENTS:

Reviewer #1 (Remarks to the Author):

The authors have responded to my initial comments and questions. I have no further issues with the manuscript.

Reviewer #2 (Remarks to the Author):

The authors have addressed all the comments well.
Pedro Cutillas

Reviewer #4 (Remarks to the Author):

In the manuscript, Dr. Huang and colleagues present an enormous amount of work including phosphoproteomics, proteomics, RNA sequencing and in vivo and ex vivo studies utilizing mouse models and human primary cells. It is without a doubt that these datasets generated on MM cell lines under different treatments, will receive good interest in the scientific community; in particular, the second half of the paper that focused on impact of spliceosome inhibition on viability of tumor cells is consistent between different MM cell lines, thus represent a promising option for combination therapy in MM patients.

However, the revised mechanistic part of the paper, despite huge effort on studying different splicing factors that might associate with splicing events in response to Cfz and Melphalan, still is not able to answer the reviewer questions. Introduction of various genes and treatment conditions did not help in clarifying and emphasizing the authors' messages.

In addition, figures were not in the order of the main text and thus further increased the complexity and confusion of the text.

As mentioned, this study has great potential and further polishing of the rational and flow of the manuscript would help in better understanding of the authors findings."

REVIEWERS' COMMENTS:

Reviewer #1 (Remarks to the Author):

The authors have responded to my initial comments and questions. I have no further issues with the manuscript.

Reviewer #2 (Remarks to the Author):

The authors have addressed all the comments well.

Pedro Cutillas

Response: We appreciate the very constructive comments of Reviewers 1 and 2 on the prior round of review and are happy to hear they find the manuscript acceptable for publication.

Reviewer #4 (Remarks to the Author):

In the manuscript, Dr. Huang and colleagues present an enormous amount of work including phosphoproteomics, proteomics, RNA sequencing and in vivo and ex vivo studies utilizing mouse models and human primary cells. It is without a doubt that these datasets generated on MM cell lines under different treatments, will receive good interest in the scientific community; in particular, the second half of the paper that focused on impact of spliceosome inhibition on viability of tumor cells is consistent between different MM cell lines, thus represent a promising option for combo-therapy in MM patients.

However, the revised mechanistic part of the paper, despite huge effort on studying different splicing factors that might associate with splicing events in response to Cfz and Melphalan, still is not able to answer the reviewer questions. Introduction of various genes and treatment conditions did not help in clarifying and emphasizing the authors' messages.

In addition, figures were not in the order of the main text and thus further increased the complexity and confusion of the text.

As mentioned, this study has great potential and further polishing of the rational and flow of the manuscript would help in better understanding of the authors findings."

Response: We acknowledge that we also were hoping to elucidate a straightforward mechanism linking PI treatment with splicing alteration in response to the prior reviewer comments. As stated previously, fully making these connections will clearly require additional work beyond the scope of this manuscript. However, we also agree with the reviewer that the large amount of additional data obtained during the revision process has made the manuscript more unwieldy. In our significant edits of the main text to conform to the desired word limits, we have done our best to improve clarity and flow of the paper. We have specifically focused on and corrected cases where references to Supplementary figures were not in a logical order. We appreciate these comments from the reviewer to motivate us to improve the manuscript.